# Quantifying the impact of aerosol scattering on the retrieval of methane from airborne remote sensing measurements

Yunxia Huang[1,2], Vijay Natraj[3], Zhaocheng Zeng[2,4], Pushkar Kopparla[5], and Yuk L. Yung[2,3]

[1]School of Science, Nantong University, Nantong, 226007, China

[2]Division of Geological and Planetary Sciences, California Institute of Technology, Pasadena, CA 91125, USA

[3]Jet Propulsion Laboratory, California Institute of Technology, Pasadena, CA 91109, USA

[4]Joint Institute for Regional Earth System Science and Engineering, University of California, Los Angeles, CA 90095, USA

[5]Graduate School of Frontier Sciences, The University of Tokyo, Kashiwa, Chiba 277-0882, Japan

*Correspondence to*: Vijay Natraj (vijay.natraj@jpl.nasa.gov)

**Abstract.** As a greenhouse gas with strong global warming potential, atmospheric methane ($CH_4$) emissions have attracted a great deal of attention. Although remote sensing measurements can provide information about $CH_4$ sources and emissions, accurate retrieval is challenging due to the influence of atmospheric aerosol scattering. In this study, imaging spectroscopic measurements from the Airborne Visible/Infrared Imaging Spectrometer–Next Generation (AVIRIS-NG) in the short-wave infrared are used to compare two retrieval techniques — the traditional Matched Filter (MF) method and the Optimal Estimation (OE) method, which is a popular approach for trace gas retrievals. Using a numerically efficient two-stream-exact-single-scattering radiative transfer model, we also simulate AVIRIS-NG measurements for different scenarios and quantify the impact of aerosol scattering in the two retrieval schemes by including aerosols in the simulations but not in the retrievals. The presence of aerosols causes an underestimation of $CH_4$ in both the MF and OE retrievals; the biases increase with increasing surface albedo and aerosol optical depth (AOD). Aerosol types with high single scattering albedo and low asymmetry parameter (such as water soluble aerosols) induce large biases in the retrieval. When scattering effects are neglected, the MF method exhibits lower fractional retrieval bias compared to the OE method at high $CH_4$ concentrations (2–5 times typical background values), and is suitable for detecting strong $CH_4$ emissions. For an AOD value of 0.3, the fractional biases of the MF retrievals are

between 1.3 and 4.5%, while the corresponding values for OE retrievals are in the 2.8–5.6% range. On
the other hand, the OE method is an optimal technique for diffuse sources (<1.5 times typical background
values), showing up to five times smaller fractional retrieval bias (8.6%) than the MF method (42.6%)
for the same AOD scenario. However, when aerosol scattering is significant, the OE method is superior
since it provides a means to reduce biases by simultaneously retrieving AOD, surface albedo and $CH_4$.
The results indicate that, while the MF method is good for plume detection, the OE method should be
employed to quantify $CH_4$ concentrations, especially in the presence of aerosol scattering.

## 1 Introduction

Atmospheric methane ($CH_4$) is about 85 times more potent per unit mass at warming the Earth than carbon dioxide ($CO_2$) on a 20-year timescale (IPCC, 2013), implying that reduction in $CH_4$ emissions could be very efficient to slow down global warming in the near term. Global mean $CH_4$ concentrations have increased from ~700 ppb in the preindustrial era to more than 1860 ppb as of 2019 (NOAA, 2019). The most effective sink of atmospheric $CH_4$ is the hydroxyl radical (OH) in the troposphere. $CH_4$ reacts with OH to reduce the oxidizing capacity of the atmosphere and generate tropospheric ozone. Increasing emissions of $CH_4$ reduce the concentration of OH in the atmosphere. With less OH to react with, the lifespan of $CH_4$ could also increase, resulting in greater $CH_4$ concentrations (Holmes et al., 2013). Soils also act as a major sink for atmospheric methane through the methanotrophic bacteria that reside within them.

Significant natural $CH_4$ sources include wetlands (Bubier et al., 1994, Macdonald et al., 1998; Gedney et al., 2004), geological seeps (Kvenvolden and Rogers, 2005; Etiope et al., 2009), ruminant animals, and termites. In addition, increased surface and ocean temperatures associated with global warming may increase $CH_4$ emissions from melting permafrost (Woodwell et al., 1998; Walter et al., 2006; Schaefer et al., 2014, Schuur et al., 2015) and methane hydrate destabilization (Kvenvolden, 1988; Archer, 2007). Human activity also contributes significantly to the total $CH_4$ emissions. Rice agriculture is one of the most important anthropogenic sources of $CH_4$ (Herrero et al., 2016; Schaefer et al., 2016). Other sources include landfills (Themelis and Ulloa, 2007), wastewater treatment, biomass burning, and methane slip from gas engines. Global fugitive $CH_4$ emissions from coal mining (Kort et al., 2014), natural gas and oil systems (Alvarez et al., 2018), hydraulic fracturing ("fracking") of shale gas wells (Howarth et al., 2011; Howarth, 2015, 2019), and residential and commercial natural gas distribution sectors (He et al., 2019) are also of increasing concern. Although the sources and sinks of methane are reasonably well known, there are large uncertainties in their relative amounts and in the partitioning between natural and anthropogenic contributions (Nisbet et al., 2014, 2016). This uncertainty is exemplified by the $CH_4$ "hiatus", which refers to the observed stabilization of atmospheric $CH_4$ concentrations from 1999–2006, and the renewed rise thereafter (Kirschke et al., 2013).

Satellite monitoring of $CH_4$ can be broadly divided into three categories: solar backscatter, thermal emission and lidar (Jacob et al., 2016). The first solar backscattering mission was SCIAMACHY (Frankenberg et al., 2006), which was operational from 2003–2012 and observed the entire planet once every seven days. It was followed by GOSAT in 2009 (Kuze et al., 2016), and subsequently the next generation GOSAT-2 in 2018 (Glumb et al., 2014). In between, the TROPOMI mission was also launched in 2017, which observes the planet once daily with a high spatial resolution of 7×7 km$^2$ (Butz

et al., 2012; Veefkind et al., 2012). CarbonSat (Buchwitz et al., 2013) is another proposed mission to
measure $CH_4$ globally from solar backscatter with a very fine spatial resolution ($2\times2$ km$^2$) and high
precision (0.4%). GHGSat-D (McKeever et al., 2017; Varon et al., 2019; Jervis et al., 2020) measures
between 1630–1675 nm, with an effective pixel resolution of $50\times50$ m$^2$ over targeted $12\times12$ km$^2$ scenes,
and is intended to detect $CH_4$ emissions from individual industrial sites. In contrast, MethaneSAT (Wofsy
and Hamburg, 2019) has a pixel size of 1–2 km$^2$ and a wide field of view (200 km$^2$) and can quantify
diffuse $CH_4$ emission sources over large areas. Thermal infrared observations of $CH_4$ are available from
the IMG (Clerbaux et al., 2003), AIRS (Xiong et al., 2008), TES (Worden et al., 2012), IASI (Xiong et
al., 2013), and CrIS (Gambacorta et al., 2016) instruments. These instruments provide day/night
measurements at spatial resolutions ranging from $5\times8$ km$^2$ (TES) to $45\times45$ km$^2$ (AIRS). GEO-CAPE
(Fishman et al., 2012), GeoFTS (Xi et al., 2015), G3E (Butz et al., 2015), and GeoCarb (Polonsky et al.,
2014) are proposed geostationary instruments (GeoCarb was selected by NASA under the Earth Venture
- Mission program), which when operational will have resolutions of 2–5 km over regional scales. The
MERLIN lidar instrument (Kiemle et al., 2014) scheduled for launch in 2021 will measure $CH_4$ by
employing a differential absorption lidar.

89         By combining a large number of footprints and high spatial resolution, airborne imaging

spectrometers are also well suited for mapping local $CH_4$ plumes. The Airborne Visible/Infrared Imaging
Spectrometer–Next Generation (AVIRIS-NG) measures reflected solar radiance across more than 400
channels between 380 and 2500 nm (Green et al., 1998; Thompson et al., 2015). Strong $CH_4$ absorption
features present between 2100 and 2500 nm can be observed at a spectral resolution of 5 nm full width
at half maximum (FWHM). A number of approaches have been developed to retrieve $CH_4$ from such
hyperspectral data. Roberts et al. (2010) used a spectral residual approach between 2000 and 2500 nm
and Bradley et al. (2011) employed a band ratio technique using the 2298 nm $CH_4$ absorption band and
2058 nm $CO_2$ absorption band. However, these techniques are not suited for terrestrial locations that
have lower albedos and have spectral structure in the SWIR. A cluster-tuned matched filter technique
was demonstrated to be capable of mapping $CH_4$ plumes from marine and terrestrial sources (Thorpe et
al., 2013) as well as $CO_2$ from power plants (Dennison et al., 2013); however, this method does not
directly quantify gas concentrations. Frankenberg et al. (2005) developed an iterative maximum *a*
*posteriori* differential optical absorption spectroscopy (IMAP-DOAS) algorithm that allows for
uncertainty estimation. Thorpe et al. (2014) adapted the IMAP-DOAS algorithm for gas detection in
AVIRIS imagery. In addition, they developed a hybrid approach using singular value decomposition and
IMAP-DOAS as a complementary method of quantifying gas concentrations within complex AVIRIS
scenes.
Accurate assessment of $CH_4$ emissions is particularly challenging in the presence of aerosols
because the latter introduce uncertainties in the light path if not accounted for. In fact, $CH_4$ emissions are
frequently correlated with pollution due to concurrent aerosol emissions. For large aerosols (such as dust),
the low Ångström exponent values result in high aerosol optical depth (AOD) values even in the
wavelength range from 2000 nm to 2500 nm (Seinfeld and Pandis, 2006; Zhang et al., 2015). Therefore,
it is important to obtain a clear understanding of aerosol impacts on $CH_4$ retrievals. In this study, SWIR
AVIRIS-NG measurements are used to analyze the impact of aerosol scattering on $CH_4$ retrievals.
Further, using an accurate but numerically efficient radiative transfer (RT) model (Spurr and Natraj,
2011), we simulate AVIRIS-NG measurements with varying aerosol amounts and quantify the impact of
aerosol scattering using two retrieval techniques, the traditional matched filter (MF) method and the
optimal estimation (OE) method that is widely used in trace gas remote sensing. This article is organized
as follows. The MF and OE retrieval methods are described in Section 2. Section 3 focuses on analysis
of a sample $CH_4$ plume detected by AVIRIS-NG measurements and compares retrievals using the MF
and OE methods. Section 4 presents a detailed evaluation of aerosol impacts on the two retrieval methods
through simulations of AVIRIS-NG spectra for different geophysical parameters. Section 5 provides a
summary of the work and discusses future research.

**2 Methods**
**2.1 MF method**

126        Real-time remote detection using AVIRIS-NG measurements are traditionally based on the MF
method (Frankenberg et al., 2016). In this method, the background spectra are assumed to be distributed
as a multivariate Gaussian $\mathcal{N}$ with covariance matrix $\Sigma$ and background mean radiance $\mu$. If $H_0$ is a
scenario without $CH_4$ enhancement and $H_1$ is one with $CH_4$ enhancement, the MF approach is equivalent
to a hypothesis test between the two scenarios:
$$H_0: L_m \sim \mathcal{N}(\mu, \Sigma) \tag{1}$$
$$H_1: L_m \sim \mathcal{N}(\mu + t\alpha, \Sigma) \tag{2}$$
where $L_m$ is the measurement radiance; $t$ is the target signature, which is defined in Equation (4); $\alpha$ is the
enhancement value, denoting a scaling factor for the target signature that perturbs the background $\mu$. If
$x$ is a vector of measurement spectra with one element per wavelength, $\alpha(x)$ can be written, based on
maximum likelihood estimates (Manolakis et al., 2014), as follows:
$$\alpha(x) = \frac{(x - \mu)^T \Sigma^{-1} t}{t^T \Sigma^{-1} t} \tag{3}$$
We utilize the same definitions as in Frankenberg et al. (2016). Specifically, the enhancement value $\alpha(\boldsymbol{x})$
denotes the thickness and concentration within a volume of equivalent absorption, and has units of ppm
× m. The target signature $\boldsymbol{t}$ refers to the derivative of the change in measured radiance with respect to a
change in absorption path length due to an optically thin absorbing layer of CH4. Note that this definition
has the disadvantage that the accuracy of the result degrades when the absorption is strong and further
attenuation becomes nonlinear. At a particular wavelength $\lambda$, $\boldsymbol{t}$ can be expressed as:
$$\boldsymbol{t}(\lambda) = -\kappa(\lambda)\boldsymbol{\mu}(\lambda), \qquad (4)$$

where $\kappa$ is the absorption coefficient for a near-surface plume with units of $\mathrm{ppm^{-1}\,m^{-1}}$. This is
different from the units of $\mathrm{m^2 \cdot mol^{-1}}$ traditionally used for the absorption coefficient $\kappa_{trad}$ in trace
gas remote sensing. Using the ideal gas law to express the volume $V$ (in liters) occupied by one mole of
CH4 at the temperature and pressure corresponding to the plume altitude ($V$ = 22.4 at standard
temperature and pressure), and the relations 1 liter = $10^{-3}\,\mathrm{m^3}$ and 1 ppm = $10^{-6}$, we obtain the
following expression for unit conversion (units in parentheses):
$\quad \kappa_{trad}\,[\mathrm{m^2 \cdot mol^{-1}}] = \kappa\,[\mathrm{ppm^{-1}m^{-1}}] \times V\,[\mathrm{liter\,mol^{-1}}] \times 10^{-3}\,[\mathrm{m^3\,liter^{-1}}]\,/\,10^{-6}\,[\mathrm{ppm^{-1}}]$ (5)
Figure 1 shows the target signature, which is calculated based on HITRAN absorption cross-sections
(Rothman et al., 2009). The background mean radiance $\boldsymbol{\mu}$ used in Equation 4 is based on the AVIRIS-
NG measurement shown in Figure 2; this is described in more detail in Section 3.
**2.2 OE method**
The OE method is widely used for the remote sensing retrieval of satellite measurements, such as
from the Orbiting Carbon Observatory-2 (OCO-2; O'Dell et al., 2018), the Spinning Enhanced Visible
and Infra-Red Imager (SEVIRI; Merchant et al., 2013), and the Greenhouse Gases Observing Satellite
(GOSAT; Yoshida et al., 2013). It combines an explicit (typically nonlinear) forward model of the
atmospheric state, a (typically Gaussian) prior probability distribution for the variabilities and a (typically
Gaussian) distribution for the spectral measurement errors. In addition, the Bayesian framework used by
the OE approach allows new information (from measurements) to be combined with existing information
(e.g., from models). In many applications, the forward model is nonlinear, and obtaining the optimal
solution requires iterative techniques such as the Levenberg–Marquardt method (Rodgers, 2000), which
has been routinely applied to study the impacts of measurement parameters on the retrieval process (see,
e.g., Zhang et al., 2015). The iteration in this algorithm follows the below procedure.
$$\mathbf{x_{i+1}} = \mathbf{x_i} + [(1+\gamma)\mathbf{S_a^{-1}} + \mathbf{K_i^T S_\epsilon^{-1} K_i}]^{-1}\{\mathbf{K_i^T S_\epsilon^{-1}}[\mathbf{y} - \mathbf{F(x_i)}] - \mathbf{S_a^{-1}}[\mathbf{x_i} - \mathbf{x_a}]\} \qquad (6)$$

where $\mathbf{x}$ is a state vector of surface and atmospheric properties, $\mathbf{S_a}$ is the *a priori* covariance matrix, $\mathbf{S_\epsilon}$
is the spectral radiance noise covariance matrix, $\mathbf{K}$ is the Jacobian matrix, $\mathbf{x_a}$ is the *a priori* state vector
and $\gamma$ is a parameter determining the size of each iteration step. The measured spectral radiance is denoted
as $y$; $F(x)$ is the simulated radiance obtained from the forward model. For the retrieval of $CH_4$ from
AVIRIS-NG measurements, the state vector includes the total column amounts of $CH_4$ and $H_2O$, while
for the retrievals from synthetic spectra, the $H_2O$ concentration is fixed and the state vector only includes
the $CH_4$ total column. The *a priori* values are within 10% of the true values; *a priori* errors are assumed
to be 20% for all state vector elements. The retrieved results are shown as the column averaged mixing
ratio ($XCH_4$, in ppm). Aerosols are not included in the state vector for both the real and synthetic
retrievals. They are, however, considered in the forward model for the synthetic simulations. Table 1
(WCRP, 1986) lists optical properties for four basic aerosol types (dust, water soluble, oceanic and soot).
Table 2 (WCRP, 1986) shows the corresponding properties for three aerosol models that are defined as
mixtures of the basic components from Table 1. We employ the Henyey-Greenstein phase function
(Henyey and Greenstein, 1941), where aerosol composition is determined by two parameters: single
scattering albedo (SSA) and asymmetry parameter ($g$). The surface albedo is also not retrieved; for both
real and synthetic retrievals, it is held fixed and assumed to be independent of wavelength.

**3 Detection and retrieval of $CH_4$ from AVIRIS-NG measurements**

To illustrate the OE retrieval and its difference from the MF method, we perform retrievals for an
AVIRIS-NG measurement made on 4 September 2014 (ang20140904t204546) in Bakersfield, CA, as
shown in Figure 2. The location is to the west of the Kern Front Oil field. This detection is a case study
from the NASA/ESA $CO_2$ and MEthane eXperiment (COMEX) campaign in California during June and
August/September 2014, which includes airborne *in situ*, airborne non-imaging remote sensing, and
ground-based *in situ* instruments to provide a real-time remote detection and measurement for $CH_4$
plumes released from anthropogenic sources. An RGB image of flight data is displayed in Figure 2a; the
emission source is a pump jack, as described in Thompson et al. (2015). Figure 2b presents results from
the MF method, which shows that the $CH_4$ plume disperses downwind and has a maximum enhancement
value of about 2800 ppm × m. Some artifacts caused by surfaces with strong absorption in the 2100–
2500 nm wavelength range, such as oil-based paints or roofs with calcite as a component (Thorpe et al.,
2013), also produce large $\alpha$ values in the MF method; these can be removed by an optimization method
such as the column-wise MF technique (Thompson et al., 2015).
Figure 3 displays the measured radiance (a) before normalization and (b) after normalization,
corresponding to two detector elements (in plume and out of plume). Every element is a cross-track
spatial location. The normalization is done by calculating the ratio of the radiance to the maximum value
across the spectral range, such that the values fall between 0 and 1. This is a first order correction for the
effects of surface albedo. Comparing the measured spectrum in plume to that out of plume, there is
obvious enhancement of $CH_4$ that is particularly evident in the normalized radiance. $CH_4$ is the main
absorber in the 2100–2500 nm wavelength range, and $H_2O$ is the major interfering gas. Figure 3b
indicates the absorption peaks due to $H_2O$ and $CH_4$.
We choose the plume center with 500 elements to illustrate results obtained using the MF and OE
methods. The former evaluates the $CH_4$ $\alpha$ value compared to the background $CH_4$ concentration, while
the latter retrieves $XCH_4$. In the MF method, the background covariance matrix $\Sigma$ and mean radiance
$\mu$ are drawn from a reference region close to the $CH_4$ emission source. These are shown in Figure 2,
where the dashed green box denotes the reference region and the source is located within the solid red
box. In the OE method, results are shown as a multiplicative scaling factor compared to a typical $XCH_4$
background of 1.822 ppm. This value is the globally averaged marine surface annual mean for 2014 (Ed
Dlugokencky, NOAA/GML, www.esrl.noaa.gov/gmd/ccgg/trends_ch4/), the year corresponding to the
AVIRIS-NG measurement being studied. We use an accurate and numerically efficient two-stream-
exact-single-scattering (2S-ESS) RT model (Spurr and Natraj, 2011). This forward model is different
from a typical two-stream model in that the two-stream approximation is used only to calculate the
contribution of multiple scattering to the radiation field. Single scattering is treated in a numerically exact
manner using all moments of the phase function. This model has been used for remote sensing of
greenhouse gases and aerosols (Xi et al., 2015; Zhang et al., 2015, 2016; Zeng et al., 2017, 2018).
Aerosols are neither included in the forward model nor retrieved in this analysis. The surface albedo is
set to a wavelength-independent value of 0.5.
Results from the two retrieval methods reveal a similar $CH_4$ plume shape (Figure 4), especially for
elements with high $CH_4$ enhancement. However, larger differences in $CH_4$ concentrations are evident in
the OE retrievals (Figure 4b). Since radiance normalization reduces the impact of surface albedo and
aerosols are not included in either retrieval, this might be due to the fact that, in the OE method, $H_2O$ and
$CH_4$ are simultaneously retrieved; the $CH_4$ retrieval has added uncertainty due to overlapping absorption
features between these two gases. The large maximum value of about 3000 in the MF method also
contributes to a reduction in relative contrast. While these results provide heuristic information about the
relative performance of the two retrieval techniques, it is difficult to compare the $CH_4$ enhancement
directly between the two methods since the background $CH_4$ concentration used in the MF method cannot
be quantified exactly. Further, evaluating retrieval biases due to ignoring aerosol scattering is not trivial
when real measurements are used. Therefore, we simulate synthetic spectra (see section 4) using the 2S-
ESS RT model to study the impacts of aerosol scattering as a function of different geophysical parameters
by varying them in a systematic manner.

**4 Aerosol impact analysis**

**4.1 Synthetic spectra**

In a real AVIRIS-NG observation, the exact column concentration of $CH_4$ cannot be controlled. However, synthetic simulations allow us to manipulate parameters such as $CH_4$ concentration, surface albedo, AOD, $g$, and SSA, and thereby test aerosol impacts on $CH_4$ retrievals. The 2S-ESS RT model is used to simulate the AVIRIS-NG spectral radiance. In this model, a prior atmospheric profile with 70 layers from the surface up to 70 km is derived from National Center for Environmental Prediction reanalysis data (Kalnay et al., 1996); absorption coefficients for all relevant gases are obtained from the HITRAN database (Rothman et al., 2009). Monochromatic RT calculations are performed at a spectral resolution of 0.5 cm$^{-1}$; the radiance spectrum is then convolved using a Gaussian instrument line shape function with a wavelength-dependent full width at half maximum (FWHM) from a calibrated AVIRIS-NG data file. The signal to noise ratio (SNR) is set to be 300, with Gaussian white noise added. This procedure results in a wavelength grid with a resolution of about 5 nm. The spectral wavelength range used to retrieve $CH_4$ is from 2100 nm to 2500 nm.

The additional atmospheric and geometric variables included in the model are listed in Table 3, which are held constant unless otherwise mentioned. The observation geometry parameters are taken from a real AVIRIS-NG measurement. Recent AVIRIS-NG fight campaigns have sensor heights ranging from 0.43 to 3.8 km; we choose a value of 1 km, the same as the highest level where aerosol is present in our simulations. The influence of AOD on $CH_4$ retrieval as a function of SSA and $g$ is analyzed in Section 4.3; in all other cases, SSA and $g$ are held constant at 0.95 and 0.75, respectively, which is representative of aerosols in the Los Angeles region (Zhang et al., 2015).

**4.2 Aerosol impact in the MF method**

We simulate synthetic spectra at different AOD, surface albedo and $CH_4$ concentration values, use the MF method to obtain the $CH_4$ enhancement, and compare differences in $\alpha$ between scenarios without and with aerosol. The covariance matrix and background mean radiance are calculated from a simulated zero AOD background with surface albedos from 0.1 to 0.5, and $XCH_4$ set at the typical background value of 1.822 ppm used in Section 3. Figure 5a shows the enhancement value as a function of $XCH_4$. As the $CH_4$ concentration increases, the enhancement value obtained by the MF method at first increases approximately linearly. However, the absorption changes in a nonlinear fashion with concentration, whereas the MF method applies a linear formalism to the change. Therefore, the enhancement value (which is correlated with the absorption signature) also shows a deviation from linear behavior at larger $XCH_4$. Two aerosol scenarios (AOD = 0, 0.3) are compared in Figure 5a, which reveals that the effect of aerosol loading is similar to an underestimation of $CH_4$ in the retrieval. The underestimation, which is

due to the shielding of $CH_4$ absorption below the aerosol layer and the fact that multiple scattering effects
between the aerosol and the surface are ignored, is clearly shown in Figure 5b, where the enhancement
value for fixed $CH_4$ concentration (same concentration as the background) decreases from 0 ppm × m
to −1532 ppm × m with increasing AOD. To clarify the impact of AOD at different surface albedo values,
zoomed in versions of $\alpha$ as a function of $XCH_4$ are presented in Figures 5c–f. For the AOD = 0 scenario,
the results are independent of surface albedo. This is because there are no multiple scattering effects
between the surface and the atmosphere (Rayleigh scattering is negligible in the retrieval wavelength
range) when there is no aerosol loading. For the scenarios with aerosol loading, the dispersion in the
zero-enhancement $XCH_4$ value between different surface albedos indicates that results from the MF
method are biased more at large AOD and surface albedo values (Figures 5d–f). This is a consequence
of increased multiple scattering between the aerosol layer and the surface that is not accounted for by the
retrieval algorithm. The maximum bias value is close to  − 700 ppm × m (equivalent to  − 0.06 × 1.822
ppm relative to the background concentration of 1.0 × 1.822 ppm) for an AOD of 0.3 and surface albedo
of 0.5 (Figure 5f). The implication of these results is that accurate knowledge of the surface albedo is
important for MF retrievals, especially when the aerosol loading is large.
A quantitative analysis of underestimation of $CH_4$ concentration due to aerosol scattering is
presented in Figure 6. The color bar shows the $\alpha$ bias — which is defined as the difference between the
enhancement value without aerosol (true $\alpha$ value) and that with aerosol — for different $CH_4$
concentrations, surface albedos and AODs. A positive bias means that $CH_4$ is underestimated. The $\alpha$ bias
increases with increasing surface albedo and AOD, reaching a maximum value of about 700 ppm × m
for the simulated cases. However, it is interesting that the bias decreases with increasing $CH_4$
concentration, which is different from the results obtained by the OE method (discussed in Section 4.3).
This surprising behavior is a direct consequence of the physical basis of the MF method. The rate of
increase in enhancement becomes smaller as $XCH_4$ becomes larger (Figure 5a). Therefore, at higher
$XCH_4$ values, the addition of aerosols (which has a similar effect as a reduction in $XCH_4$) results in a
lower reduction in enhancement compared to that at lower $XCH_4$ values, resulting in a net decrease in
the enhancement bias.
**4.3 Aerosol impact in the OE method**
For the simulation of the synthetic spectra, we assume nonzero aerosol loading below 1 km elevation.
The OE method is then used to perform retrievals using the same configuration (including, in particular,
the same surface albedo) except that AOD is set to zero. This approach is similar to neglecting aerosol
scattering in the $CH_4$ retrieval; the retrieval bias is defined as the difference between the true $XCH_4$ in
the simulation and the retrieved value (positive values refer to underestimation). First, we study the
retrieval bias caused by different aerosol types and mixtures. Figure 7a shows $CH_4$ retrieval biases as a
function of SSA and $g$; surface albedo and AOD are kept constant at 0.3 and $XCH_4$ is assumed to be 1.0
× 1.822 ppm. The retrieval bias increases with SSA and decreases with $g$, with a maximum bias ratio
(ratio of retrieval bias to the true value) of about 20%. This behavior can be explained as follows. At
higher SSA values, there are more multiple scattering effects (that are ignored in the retrieval). On the
other hand, larger values of $g$ imply greater anisotropy of scattering (preference for forward scattering),
leading to a reduction in multiple scattering effects. Since the retrieval bias is large for high SSA and
low $g$, the water-soluble aerosol type (Table 1) and the maritime aerosol model (Table 2) can be expected
to induce greater biases in the retrieval. In order to compare the impacts of SSA and $g$ in further detail,
retrieval results due to a ± 5% change in SSA and $g$ for the three aerosol models from Table 2 are shown
in Figures 7b and 7c. Note that for the maritime aerosol model, the SSA is set to 0.999 for the +5%
scenario to ensure physicality. It is clear that (1) the maritime aerosol model induces larger retrieval
biases than the other aerosol types, and (2) the retrieval results are more sensitive to changes in $g$ than
those in SSA.
We then simulate synthetic spectra for different values of $CH_4$ concentration, surface albedo and
AOD. The impacts of aerosol scattering on the retrievals for these scenarios are demonstrated in Figure
8. Figure 8a shows a 5 × 5 panel of boxes. Within each box, $XCH_4$ is constant, while surface albedo
increases from top to bottom and AOD increases from left to right. The variation of $XCH_4$ across the
boxes is shown in Figure 8b. We also show a zoomed in plot of the bottom right box ($XCH_4 = 5.8 \times$
1.822 ppm) in Figure 8c, which illustrates the AOD and surface albedo changes within a box. These
changes are identical for all boxes. Figure 8a indicates that OE retrievals produce larger $CH_4$ biases at
higher $XCH_4$ values, in contrast with MF results. In addition, it is evident that the retrieved $CH_4$ bias
increases with increasing AOD. The $CH_4$ bias induced by differences in the surface albedo is not as large
as that due to AOD variations, but surface albedo effects are noticeable at large AOD. Figure 8d shows
the sensitivity of retrieval biases to changes in AOD and surface albedo, again demonstrating the greater
impact of AOD than surface albedo in the retrieval.
The effects of changing the *a priori*, *a priori* error and RT simulation spectral resolution on the
retrieved $XCH_4$ are shown in Figure 9. For these calculations, the other parameters are set as follows:
SSA = 0.95, $g$ = 0.75, AOD = 1.0, surface albedo = 0.5 and true $XCH_4$ = 5.8 × 1.822ppm. The parameters
were chosen to correspond to the scenario with the largest retrieval bias in Figure 8c (bottom right box
in Figure 8c). Figure 9a shows that the retrieved $XCH_4$ changes by about 9 ppb as the *a priori* changes
from half to twice the true $XCH_4$ value. Similarly, the $XCH_4$ difference is less than 4 ppb when the *a*
*priori* error changes from 0.05 to 0.5 (Figure 9b). Compared to the bias of about 923 ppb induced by
neglecting aerosol scattering for this scenario, it is clear that the impacts of the *a priori* and *a priori* error
are very small. The effect of spectral resolution is larger, but $XCH_4$ still changes by only about 100 ppb
when the spectral resolution is changed from 0.5 to 0.1 $cm^{-1}$ (Figure 9c).
**4.4 Comparison of the two retrieval techniques**

340        Figure 10 presents the bias ratios for the two retrieval techniques at different AODs (surface albedo

= 0.3). In the MF method, the bias ratio is defined as the ratio of the bias to the true value of $\alpha$. On the
other hand, in the OE method, it is the ratio of the bias to the true $XCH_4$. From Figure 10 it is clear that
the bias ratio decreases with increasing $CH_4$ concentration and has higher values at larger AODs. The
bias ratio for the MF method (1.3–4.5%) is up to 53.6% less than that for the OE method (2.8–5.6%) for
AOD = 0.3 when the $CH_4$ concentration is high (2–5 times typical background values). On the other
hand, the OE method performs better when enhancements are small and $XCH_4$ is close to the background
value. For example, the bias ratio for the MF method has a high value of about 42.6% at AOD = 0.3 for
a 10% enhancement ($XCH_4 = 1.1 \times 1.822$ ppm); the OE value for the same scenario is 8.6%. For scenarios
where scattering is ignored, the two retrieval techniques seem to be complementary, with differing
utilities for different enhancements. On the other hand, when RT models that account for scattering
effects are employed, the MF technique is suboptimal. Further, MF retrievals rely on accurate
characterization of the surface albedo, especially when the aerosol loading is large. Finally, the MF
method does not retrieve concentrations, which are necessary to infer fluxes. Therefore, the OE technique
is in general superior due to its ability to support simultaneous retrieval of aerosols, surface albedo and
$CH_4$ concentration.

**5 Summary and discussion**

358        Remote sensing measurements from airborne and satellite instruments are widely used to detect

$CH_4$ emissions. In our study, the traditional MF and the OE methods are used to quantify the effects of
aerosol scattering on $CH_4$ retrievals based on simulations of AVIRIS-NG measurements. The results
show that the retrieval biases increase with increasing AOD and surface albedo for both techniques. In
the OE method the biases increase with increasing $CH_4$ concentration and SSA, but decrease with
increasing aerosol asymmetry parameter. The $CH_4$ retrieval bias increases with increasing $XCH_4$ in the
OE method but decreases for the same scenario in the MF method. The surprising MF trend is attributed
to the inability of the MF method to treat nonlinear absorption effects at high $XCH_4$ values. We also
present bias ratios for the two techniques. The MF method shows smaller bias ratios at large $CH_4$
concentrations than the OE method; it is, therefore, the optimal method to detect strong $CH_4$ emission
sources when scattering effects can be ignored in the retrieval. For the same retrieval scenario, the OE
method seems to be more suitable for detecting diffuse sources. Further, the MF method relies on a
comparison with the background $CH_4$ concentration. It is difficult to get an accurate estimate of the
background $XCH_4$ value in polluted atmospheric environments. In contrast, the OE method provides
retrievals based solely on the atmospheric scenario of interest; $CH_4$, aerosols and surface albedo can be
simultaneously inferred. Therefore, when scattering effects need to be considered, the OE method is the
appropriate choice. Indeed, the MF method was intended for plume detection. OE enables accurate
quantification of $XCH_4$ in the presence of aerosol scattering.
This study focused on a comparison of retrieval techniques. It is also important to accurately
represent the physics of atmospheric RT, especially for scenarios with significant aerosol scattering. RT
models traditionally used in retrievals of imaging spectroscopic data use simplified radiation schemes
and predefined aerosol models, which may introduce inaccurate in the representation of atmospheric
physics. The 2S-ESS model provides the capability to quantify aerosol impacts on $CH_4$ retrieval for
different aerosol types, optical depths and layer heights. In future work, we will compare retrievals using
the 2S-ESS model against those from other commonly used models such as MODTRAN. We will also
evaluate the impact of varying instrument spectral resolution and signal to noise ratio for simultaneous
retrieval of $CH_4$, surface albedo and AOD. This will be relevant for the design of imaging spectrometers
for planned future missions such as the NASA Surface Biology and Geology (SBG) mission.

**Data availability**
The code and data are available from the authors upon request.

**Author contributions**
VN conceived the work, provided the radiative transfer and aerosol models, supervised YH, and
assisted with manuscript preparation. YH designed and performed the retrievals, analyzed the results,
and prepared the original manuscript. ZZ contributed to retrieval setup and assisted with analysis of the
results. PK provided valuable inputs into the science of $CH_4$ remote sensing. YLY supervised YH and
participated in the evaluation of the retrieval results and intercomparison. All listed authors contributed
to the review and editing of this manuscript.

**Competing interests**
The authors declare that they have no conflict of interest.

**Acknowledgements**
A portion of this research was carried out at the Jet Propulsion Laboratory, California Institute of
Technology, under a contract with the National Aeronautics and Space Administration
(80NM0018D0004). The authors gratefully acknowledge the insightful and constructive comments from
the two anonymous reviewers, which improved the clarity and quality of the manuscript, and elevated
the significance of the work beyond the original expectation.

**Financial Support**
VN acknowledges support from the NASA "Utilization of Airborne Visible/Infrared Imaging
Spectrometer Next Generation Data from an Airborne Campaign in India" program (solicitation
NNH16ZDA001N-AVRSNG), and the Jet Propulsion Laboratory Research and Technology
Development program. PK was funded by the Japan Society for the Promotion of Science International
Research Fellow Program.

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

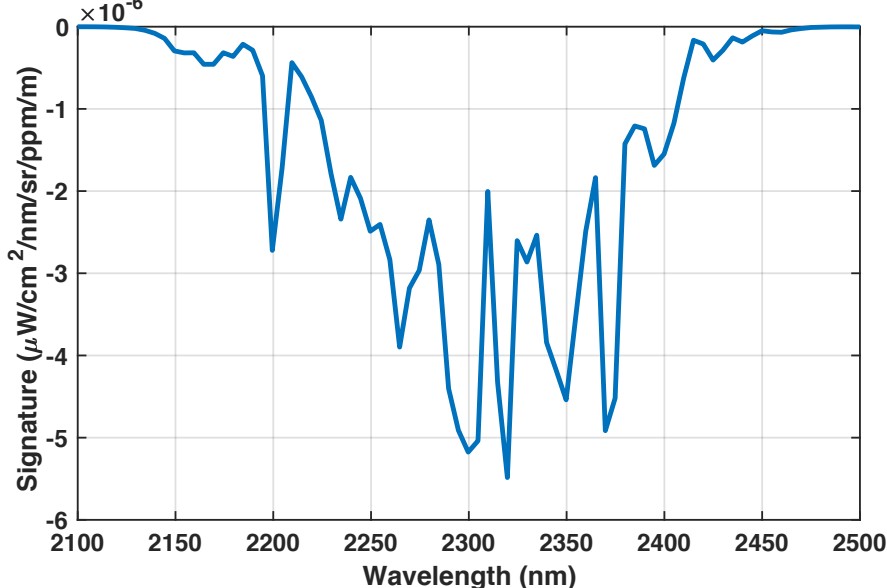



**Figure 1: The target signature used for the Matched Filter method.**


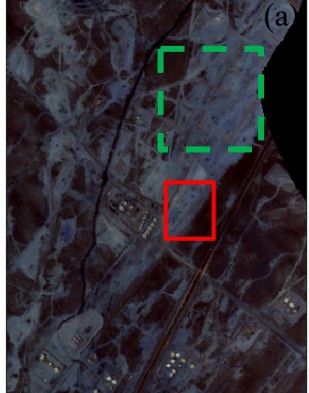 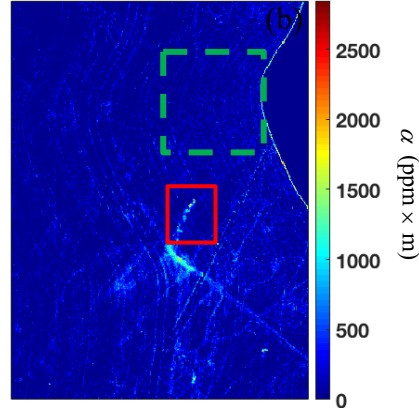


**Figure 2: (a) RGB image of flight data from 4 September 2014 (ang20140904t204546). Adapted from Thompson et al. (2015). (b) CH$_4$ enhancement value $\alpha$ (ppm × m) obtained by the MF method. An emission source is shown in the solid red box and the background region near the target for the MF calculation is indicated by the dashed green box.**







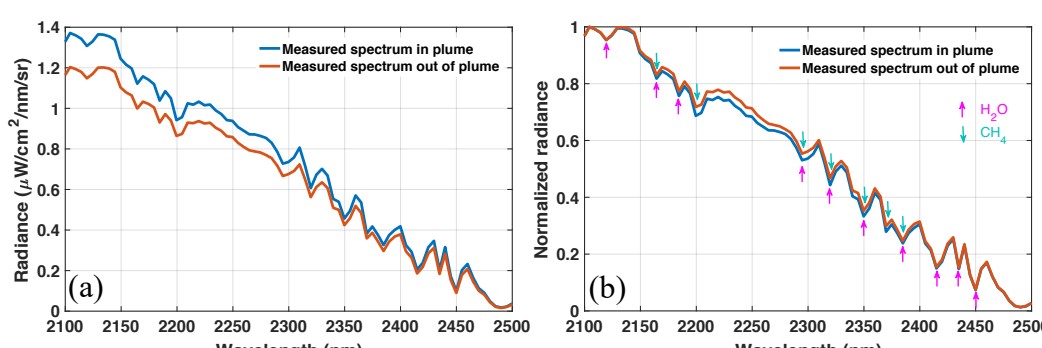


**Figure 3: (a) Real radiance and (b) normalized radiance at cross-track detector elements (in and out of plume)**

**from the sample AVIRIS-NG measurement. The colored arrows in (b) show the main absorption features due**

**to $H_2O$ (purple) and $CH_4$ (green).**



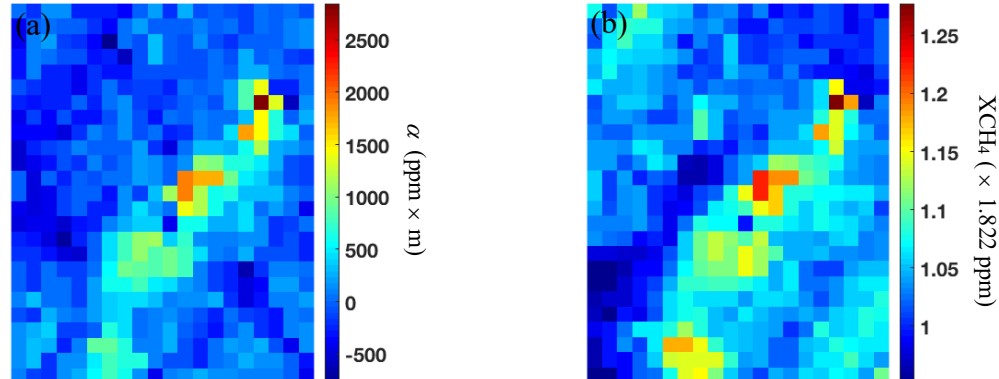


**Figure 4: Retrieval image for the plume center (500 elements) based on the (a) MF method and (b) OE method.**



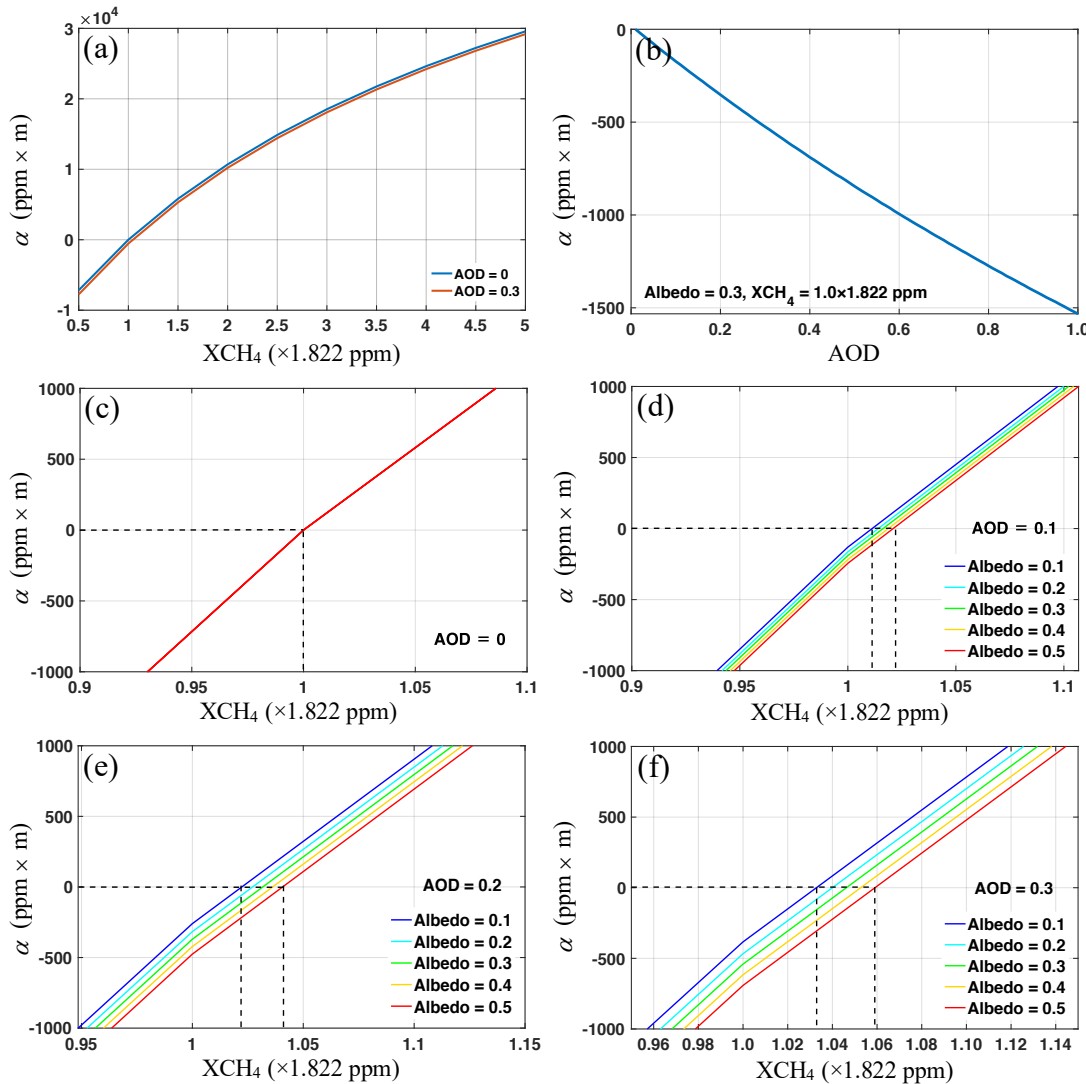

Figure 5: (a) $\alpha$ as a function of XCH$_4$ for AOD = 0 and AOD = 0.3 (surface albedo = 0.3). (b) $\alpha$ as a function of AOD (XCH$_4$ = 1.0 × 1.822 ppm, surface albedo = 0.3). Zoomed in versions of $\alpha$ as a function of XCH$_4$ for different surface albedos (0.1-0.5), where (c) AOD = 0, (d) AOD = 0.1, (e) AOD = 0.2, and (f) AOD = 0.3.



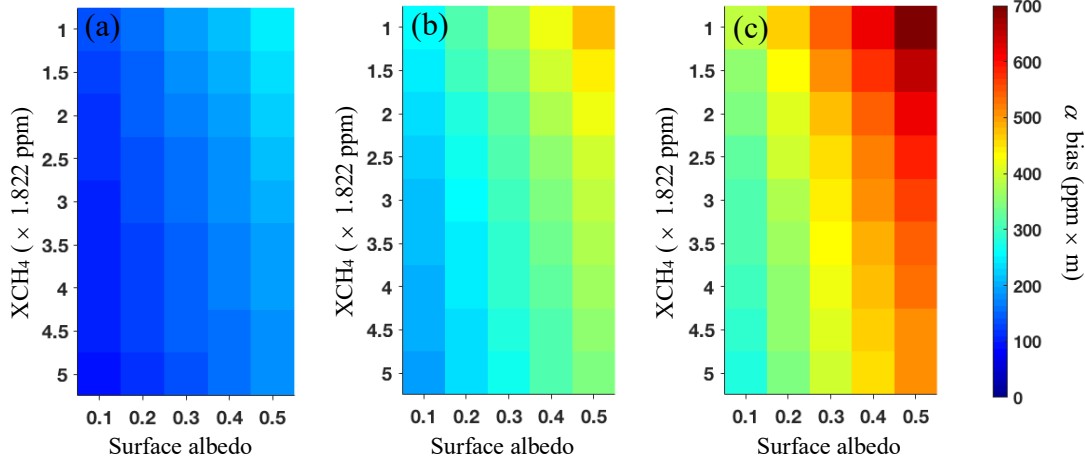


**Figure 6: Bias in $\alpha$ as a function of XCH$_4$ and surface albedo for (a) AOD = 0.1, (b) AOD = 0.2, and (c) AOD**
**= 0.3.**



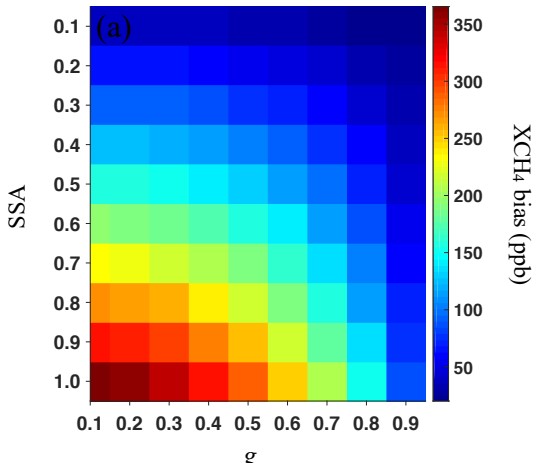



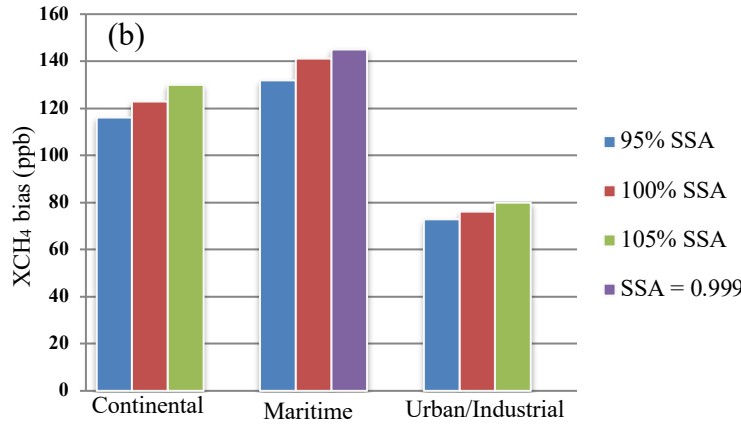


**Figure 7: (a) CH$_4$ retrieval biases for different values of *g* and SSA. Surface albedo, AOD = 0.3, XCH$_4$ = 1.0**
**× 1.822 ppm. (b) CH$_4$ retrieval biases for a ± 5% change in SSA for the three aerosol mixture models. (c) Same**
**as (b), but for a ± 5% change in *g*.**

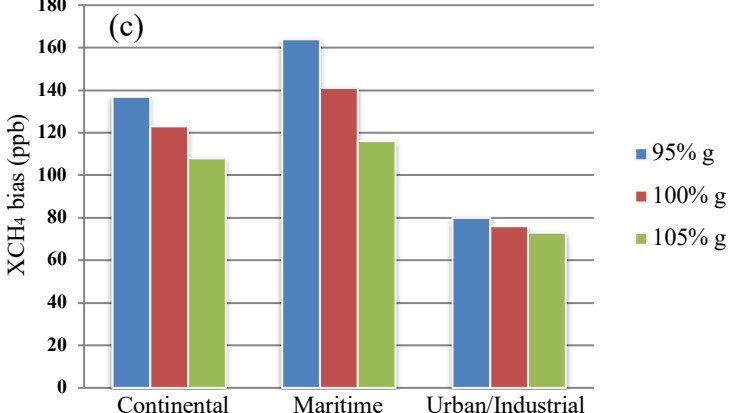

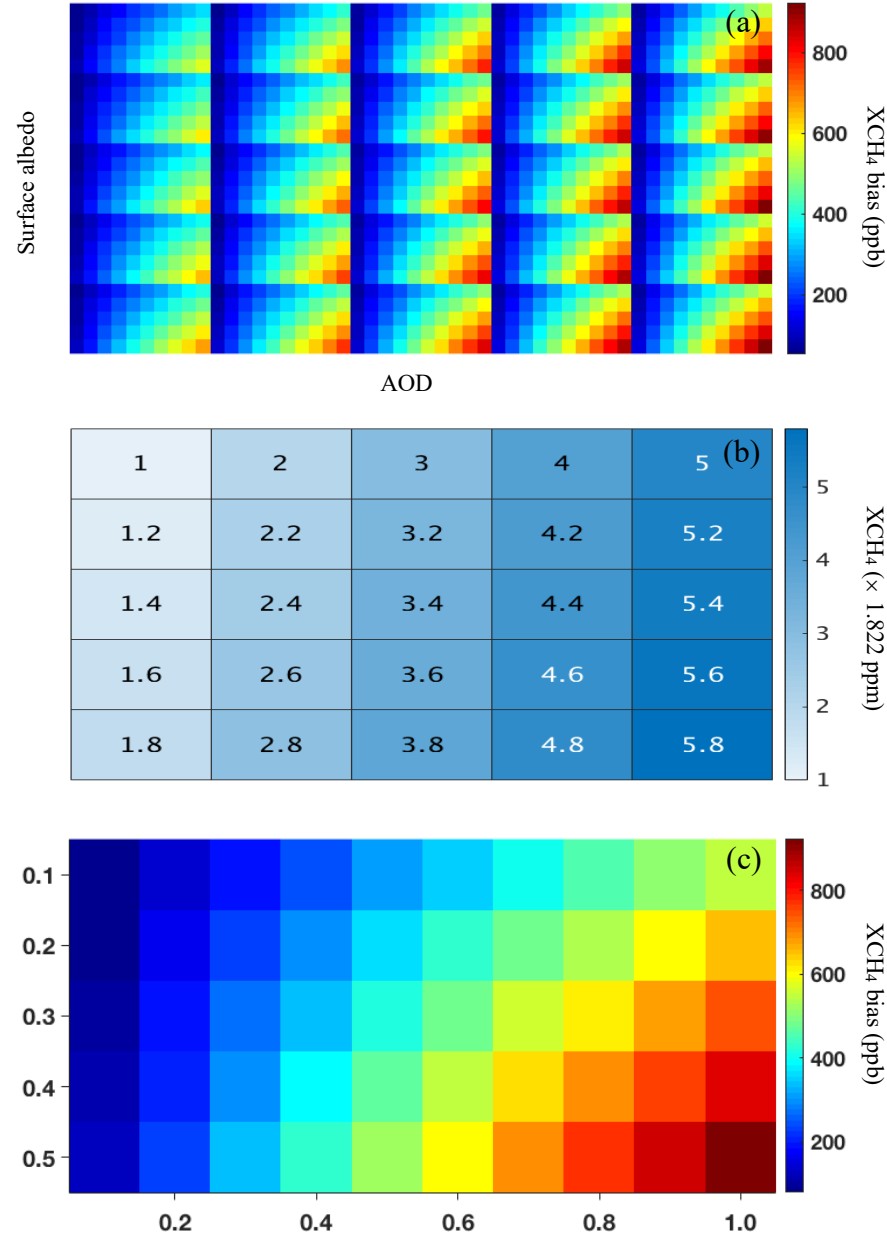

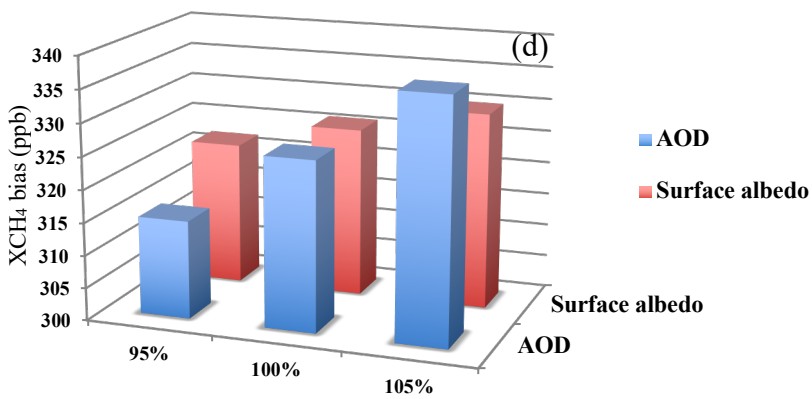


**Figure 8: (a) CH$_4$ retrieval biases for different values of XCH$_4$, AOD and surface albedo. *g* = 0.75, SSA = 0.95.**

**(b) XCH$_4$ for each box in (a). (c) Zoomed in plot of bottom right box (XCH$_4$ = 5.8 × 1.822 ppm). The x and y**

**axes show the variation of AOD and surface albedo, respectively. These changes are identical for every box**

**in (a). (d) CH$_4$ retrieval biases for a ± 5% change in AOD and surface albedo from a base value of 0.3 (*g* =**

**0.75, SSA = 0.95, XCH$_4$ = 5.8 × 1.822 ppm).**



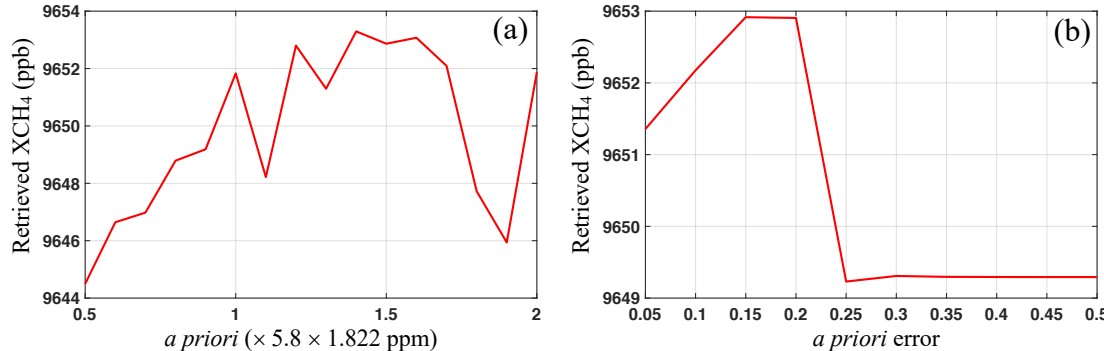



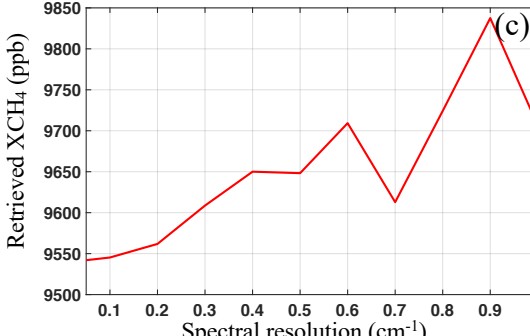


**Figure 9: Retrieved XCH₄ for different values of (a)** *a priori* **(***a priori* **error = 0.2), (b)** *a priori* **error (***a priori* **= 5.5 × 1.822 ppm) and (c) spectral resolution.** *g* **= 0.75, SSA = 0.95, AOD = 1.0, surface albedo = 0.5, XCH₄ = 5.8 × 1.822 ppm.**



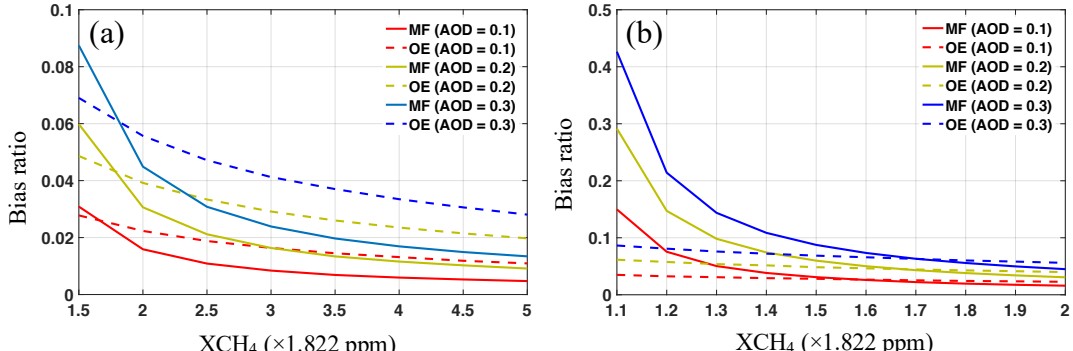

**Figure 10: (a) Bias ratio as a function of CH$_4$ concentration for the two retrieval techniques, where the XCH$_4$**
**ranges from 1.5 to 5 (× 1.822 ppm). (b) Same as (a), but for XCH$_4$ ranging from 1.1 to 2 (× 1.822 ppm). Surface**
**albedo is set to 0.3 for all cases; results for the MF and OE methods are shown by solid and dashed lines,**
**respectively.**


| | Dust-like | Water soluble | Oceanic | Soot |
|---|---|---|---|---|
| **SSA** | 0.805 | 0.799 | 0.970 | 0.014 |
| *g* | 0.926 | 0.550 | 0.816 | 0.092 |

**Table 1: Optical properties of basic aerosol types (WCRP, 1986).**

|  |  | Continental | Maritime | Urban/Industrial |
|---|---|---|---|---|
| **Aerosol component** | Dust-like | 70% |  | 17% |
| | Water soluble | 29% | 5% | 61% |
| | Oceanic |  | 95% |  |
| | Soot | 1% |  | 22% |
| **SSA** | | 0.746 | 0.966 | 0.314 |
| **g** | | 0.764 | 0.810 | 0.586 |

**Table 2: Optical properties of three aerosol mixture models (WCRP, 1986).**


| Attribute | Values |
|---|---|
| Sensor height | 1 km |
| View zenith angle | 11.91° |
| Solar zenith angle | 30.75° |
| Relative azimuth angle | 22.87° |
| Aerosol loading region | surface to 1 km |
| SSA | 0.95 |
| *g* | 0.75 |

**Table 3: Inputs for the 2S-ESS model simulation.**