# Peer review of "Quantifying the impact of aerosol scattering on the"

_Atmospheric Measurement Techniques, 2020_

## Referee Comment (RC1) · Anonymous Referee #1 · 15 Jun 2020

**General comments**

This paper provides some analysis about how aerosols properties affect  $CH_4$  retrieval, which will attract a lot of interests from the audience of this journal. However, it is suggested that more specific analysis about the aerosol model are needed and the main points about aerosol impact need to be emphasized in both abstract and main part. Moreover, in the two retrieval algorithm used in this study, no aerosol loading is included. I'm just wondering if AOD or other aerosol parameters are retrieved simultaneously with XCH4, such as adding AOD in the state vector of OE retrieval, will the retrieval bias be improved? If any preliminary results could be shown, it will be interesting. Furthermore, the section 3 has less close relationship with the topic of this paper, the authors are suggested to think it more.

**Specific comments**

- 1. In the third paragraph of Introduction, I suggest the authors to add more the description about how to retrieve CH4 concentration from satellite measurements, especially the advantage of hyperspectral imaging in CH4 retrieval. I think the description about atmospheric correction has less relationship with the topic of this paper.
- 2. Line 171-172: How to do normalization for measured radiance? Add some description about this, please.
- 3. Line 181: Is the typical XCH4 background of 1.822 ppm shown by the authors here related to the background covariance matrix and mean radiance used in MF method? Some reasons are expected here. By the way, it is better to mention the background covariance matrix and mean radiance in MF retrieval of CH4 plume case here.
- 4. In the OE retrieval in section 3, what is the definition of the a priori value of XCH4? What aerosol model do the authors use? I think some parameters about aerosol model are expected here.
- 5. In section 4.3, the authors show the variation of OE XCH4 retrieval bias with SSA, g, AOD, surface albedo and XCH4. Which parameters affect XCH4 retrieval bias most? From aerosol parameters, which type of aerosols, such as smoke, dust or sea salt, causes the largest or lowest bias in XCH4 retrieval? These information will attract the audience's interest and provide guidance to correct aerosol impact in future XCO4 retrieval algorithm.
 In OE retrieval, the a priori error of XCH4 will affect the retrieval bias as well. Maybe the authors could check its impact.

**Technical corrections**

1. Figure 9a and 9b have some overlaps with the same XCH4. There is no need to express them using two figures.

---

## Referee Comment (RC2) · Anonymous Referee #2 · 28 Jul 2020

This paper studies the impact of aerosols on methane retrievals from synthetic AVIRIS-NG-like measurements using two retrieval methods, the traditional match filter (MF) method, and the more modern and quantitative optimal estimation (OE) method that uses radiative transfer model and can include more physics. It shows how both retrievals are sensitive to various AOD, CH4 concentration, surface albedo, SSA and g. The scope of this study is well suited for AMT. This paper is generally well organized and methodology is generally good. However, a lot of results are shown without much further explanation about the physics behind. Some of the statements may be questionable due to different definitions of retrieval biases between the two retrievals. Some key information (e.g., surface albedo treatment in OE retrieval) is not clearly de-

scribed. The abstract also needs to be improved as it does not clearly summarize this study. Overall, I think that this paper can be published after addressing the specific comments below.

Specific comments:

1. Subscripts and superscripts in the text are disproportionally too small to read

2. L19-21, this sentence is not consistent with the text as real AVIRIS-NG data are mainly used to compare both MF and OE retrievals, rather than analyze the impact of aerosol scattering on CH4 retrievals. That is probably why the first reviewer commented that section 3 is loosely connected with the main purpose of this paper. I suggest clearly describing the purpose of this section 3 probably in the introduction section in relation to the main topics of this study.

3. In abstract, the sentence in L25-29, it is not clear about what kind of scenario for the retrieval bias. Please make it clear here that this for retrieval underestimation of CH4 when aerosol is present but neglected in the retrieval. The use of 50% here and also in the text in Section 4.4 is very confusing especially you have >100% enhancement and < 50% enhancement and also the retrieval bias is actually on the order of 1.5-6%. You may use something like "half of the retrieval bias" or provide specific retrieval bias ($\sim$2-6% for OE and 1,5-3.5% for MF) based on Figure 9. The sentence is also too long. You may rephrase it in a couple of sentences.

4. In abstract, L29-31, it is good to summarize main results instead of just describing what are discussed.

5. L74, suggest changing "large" to "large number of " as it implies coarser spatial resolution contrary to "fine spatial resolution"

6. L78, suggest changing to "a spectral resolution of 5 nm full width at half maximum (FWHM)"

7. At the end of the introduction, it would be useful to add how this paper is organized

in following sections

8. Units on both sides of Equation (5) do no match. According to the text, V has a unit of liter / mol or 1E-3 mˆ3/mol, and 1/(V*1E3) has a unit of mol/mˆ(-3). Maybe 1/(V*1E3) should be V*1E-3 instead. Or V has a unit of mol/liter, then it should not be called V as it is confusing. So please clarify this.

9. L132-135, the sentence does not read well here as the purpose of using real AVIRIS-NG data has not been introduced yet. You may rephrase it to something like "To illustrate the MF retrieval and its difference from the OE method, we perform MF retrievals from AVIRIS-NG measurement made on . . . as shown in Fig. 2. The samples for . . ." Or it might be even better to move these two sentences to Section 3 before showing MF retrieval results.

10. L167, it might be good to describe some of the retrieval artifacts and why they are produced. Are some of the retrieval artifacts related to aerosols or surface albedo?

11. L169-172, the first sentence seems to be redundant with previous description and can be removed. Also good to describe how the normalization is done and its main purpose.

12. L173-174, the sentence "The radiance has units .." can be removed as the spectral range has already been mentioned earlier in the paragraph and it is not necessary to mention the units of radiance.

13. L206, the spectral resolution of 0.5cmˆ-1 seems to be too coarse to resolve monochromatic spectral features in this spectral region. Have you performed sensitivity calculation to see how this affect the synthetic AVIRIS-NG radiance?

14. L178-186, it is good to mention clearly whether aerosol is included in both the forward model and retrieval. It seems that aerosol is not retrieved, but not sure if fixed aerosol model is used in the forward model as it mentions "Single scattering . . . using all moments of the phase function"

[Figure]

15. L187-191, although H2O is not retrieved and taken into account in the MF method, it should also cause retrieval bias/uncertainty to the MF result. Probably it will cause different retrieval errors to MF method and OE method due to its different retrieval treatments. Also are some of the differences due to aerosols and surface albedo?

16. L226, the absorption cross-section is independent of concentration, suggest removing "cross-section"

17. L228, why does the effect of aerosol loading cause underestimation? Would be good to provide some explanation. Due to the shielding of CH4 absorption below aerosol layer?

18. L232-233 and in Fig. 5c, are the results really independent of surface albedo here? Or is this simply because a background with the same surface albedo is used? In actual MF retrievals, surface albedo is not necessarily known (or be the same as that in the background). Also according to normalization procedure shown in Fig. 2, looks like most of the surface albedo can be taken into account after the normalization if surface albedo is not perfectly known as in real retrievals, but bias will occur. Please clarify this.

19. L236, good to explain why larger biases at large AOD and surface albedo values. Also since the enhancement in units of ppm m is retrieved with the MF method, it is better to mention the bias in enhancement (∼-700 ppm m) rather than saying "maximum bias .. close to 0.06 x 1.822 ppm), or you can say "the maximum bias is close to ∼-700 ppm m (equivalent of -0.06 x 1.822 ppm) ..."

20. L242-243, why does the bias decreases with increasing CH4 concentration for the MF method? The reason givens on L244-245 only shows that the enhancement is more underestimated at larger XCH4 concentration (as shown from the curves in Figure 5a that deviated from a straight line), and seems not able to explain the enhancement difference between without and with aerosols decreases with increasing CH4 concentration.

[Figure]

21. Figure 6, the bias is negative (underestimated) as indicated in the text. Suggest making it clear in Fig. 6 caption that the figure shows the magnitude of underestimation.

22. L255 and Figures 7,8,9, is the bias also negative? If so, please make it clear.

23. L258, good to explain how the bias varies with SSA and g.

24. L259-267 and results in Figure 8: is surface albedo retrieved? It seems to me that it is retrieved with XCH4 so that the error is small for different surface albedo when AOD=0. While for other cases (e.g., surface albedo is kept constant at 0.3), maybe surface albedo is not retrieved. Please make it clear probably at the end of section 2.2 or in this paragraph about whether surface albedo is retrieved and how it is retrieved (e.g., wavelength independent or dependent)

25. L264, it seems to me that the bias is defined differently for the MF case, as the enhancement between without and with aerosols (L239-240), while the bias for OE is defined as the difference between retrieved and true XCH4. If we use a similar definition, according to Figure 5a, there is larger underestimation at higher XCH4 values for both without and with aerosols in the MF method.

26. L273, please check if it should be between "with and without aerosols" as the case with zero AOD is the truth reference for the MF method.

27. L275-281, how is this OE retrieval sensitive to the assumed a priori error of 20%? If you use a larger a priori error for the OE method, will the conclusion here be changed?

28. L280, an example is given for a XCH4 of 1.1x1.822ppm. It is useful to give another example at high XCH4, for example XCH4=5.0.

29. Section 5 is a summary of this paper and discussion about future work, I suggest changing this section title to "Summary and discussion"

30. L289-291, the sentence might not be true as mentioned earlier due to different bias definitions used for OE and MF methods.

---

## Author Comment (AC1) · 24 Aug 2020

**Dear Editor,**

We would like to thank you and the two reviewers for your constructive comments and suggestions to improve the clarity of our manuscript. We have made changes to address these comments and suggestions. The following are the main changes:

- 1. More information has been added to the abstract to increase clarity, quantify the results better and summarize the comparison between the MF and OE methods.
- 2. Details have been added about the treatment of aerosols and surface albedo.
- 3. Explanations are supplied for the behavior of the retrievals as a function of the different parameters.
- 4. More quantification is provided for the differences between the two retrieval techniques.
- 5. Effects of changing the *a priori*, *a priori* error and simulation spectral resolution are described.
- 6. New Tables and Figures have been added to provide more detail.

Point-by-point responses to the comments are provided below. The reviewer comments are in blue, our responses are in red (line numbers refer to those in the revised manuscript), and modifications to the original manuscript are highlighted in yellow.

**Vijay Natraj On behalf of all co-authors**

This paper provides some analysis about how aerosols properties affect CH4 retrieval, which will attract a lot of interests from the audience of this journal. However, it is suggested that more specific analysis about the aerosol model are needed and the main points about aerosol impact need to be emphasized in both abstract and main part.

We thank the reviewer for the excellent suggestion. We have added some sentences (lines 27–30) in the abstract.

The presence of aerosols causes an underestimation of CH4 in both the MF and OE retrievals; the biases increase with increasing surface albedo and aerosol optical depth (AOD). Aerosol types with high single scattering albedo and low asymmetry parameter (such as water soluble aerosols) induce large biases in the retrieval.

We added Tables 1 and 2 and provided a description of the aerosol models (lines 173–178).

Table 1 lists optical properties for four basic aerosol types (dust, water soluble, oceanic and soot). Table 2 shows the corresponding properties for three aerosol models that are defined as mixtures of the basic components from Table 1 (WCRP, 1986). We employ the

Henyey-Greenstein phase function (Henyey and Greenstein, 1941), where aerosol composition is determined by two parameters: single scattering albedo (SSA) and asymmetry parameter (g).

We also added more description of the aerosol impact in the main text (lines 305–312). Further, we added Figures 7b and 7c.

Since the retrieval bias is large for high SSA and low g, the water-soluble aerosol type (Table 1) and the maritime aerosol model (Table 2) can be expected to induce greater biases in the retrieval. In order to compare the impacts of SSA and g in further detail, retrieval results due to a  $\pm$  5% change in SSA and g for the three aerosol models from Table 2 are shown in Figures 7b and 7c. Note that for the maritime aerosol model, the SSA is set to 0.999 for the +5% scenario to ensure physicality. It is clear that (1) the maritime aerosol model induces larger retrieval biases than the other aerosol types, and (2) the retrieval results are more sensitive to changes in g than those in SSA.

Moreover, in the two retrieval algorithm used in this study, no aerosol loading is included. I'm just wondering if AOD or other aerosol parameters are retrieved simultaneously with XCH4, such as adding AOD in the state vector of OE retrieval, will the retrieval bias be improved? If any preliminary results could be shown, it will be interesting.

We appreciate the reviewer's suggestion. The issue is that the MF method does not permit retrieval of AOD; it has traditionally been intended to provide a quick detection of CH4. The OE method, on the other hand, is more flexible and does allow aerosol retrieval. We did not add AOD to the state vector since one of the methods was incapable of handling it, and we would not be able to make a meaningful comparison. In this work, we instead study the aerosol impact indirectly, by including it in the simulations but not in the retrieval. Through this process, we demonstrate that the MF method has larger biases for diffuse sources. We indicate in the abstract (lines 24–27) that the AOD is not included in the state vector.

Using a numerically efficient two-stream-exact-single-scattering radiative transfer model, we also simulate AVIRIS-NG measurements for different scenarios and quantify the impact of aerosol scattering in the two retrieval schemes by including aerosols in the simulations but not in the retrievals.

The reviewer makes an important point, though. We modify/add the following sentences (lines 344–351) indicating that the MF method is also not optimal for scenarios with aerosol scattering.

For scenarios where scattering is ignored, the two retrieval techniques seem to be complementary, with differing utilities for different enhancements. On the other hand, when RT models that account for scattering are employed, the MF technique is suboptimal. Further, MF retrievals rely on accurate characterization of the surface albedo, especially when the aerosol loading is large. Finally, the MF method does not retrieve concentrations, which are necessary to infer fluxes. Therefore, the OE technique is in general superior due to its ability to support simultaneous retrieval of aerosols, surface albedo and CH4 concentration.

We also add two sentences (lines 37–40) in the abstract and add/modify some sentences (lines 362–364, 369–371) in the summary.

However, when aerosol scattering is significant, the OE method is superior since it provides a means to reduce biases by simultaneously retrieving AOD, surface albedo and CH4. The results indicate that, while the MF method is good for plume detection, the OE method should be employed to quantify CH4 concentrations, especially in the presence of aerosol scattering.

The MF method shows smaller bias ratios at large CH4 concentrations than the OE method; it is, therefore, the optimal method to detect strong CH4 emission sources when scattering effects can be ignored in the retrieval.

Therefore, when scattering effects need to be considered, the OE method is the appropriate choice. Indeed, the MF method was intended for plume detection. OE enables accurate quantification of XCH4 in the presence of aerosol scattering.

Furthermore, the section 3 has less close relationship with the topic of this paper, the authors are suggested to think it more.

We feel that this section belongs in the paper. Section 3 provides a comparison of MF and OE retrievals from a real AVIRIS-NG measurement. These results provide heuristic information about the relative performance of the two techniques. However, there are some difficulties in comparing these retrievals. Further, understanding the retrieval effects of ignoring aerosol scattering is easier when we employ simulations. Therefore, both methods of comparison are useful and illustrative. We add/modify the following sentences (lines 225–231):

While these results provide heuristic information about the relative performance of the two retrieval techniques, it is difficult to compare the CH4 enhancement directly between the two methods since the background CH4 concentration used in the MF method cannot be quantified exactly. Further, evaluating retrieval biases due to ignoring aerosol scattering is

not trivial when real measurements are used. Therefore, we simulate synthetic spectra (see section 4) using the 2S-ESS RT model to study the impacts of aerosol scattering as a function of different geophysical parameters by varying them in a systematic manner.

**Specific comments**

1. In the third paragraph of Introduction, I suggest the authors to add more the description about how to retrieve  $CH_4$  concentration from satellite measurements, especially the advantage of hyperspectral imaging in  $CH_4$  retrieval. I think the description about atmospheric correction has less relationship with the topic of this paper.

The reviewer is right. The description of atmospheric correction did not flow well with the rest of the introduction. We removed that paragraph and added a paragraph on satellite retrieval of CH4 concentrations (lines 68–84).

Satellite monitoring of CH4 can be broadly divided into three categories: solar backscatter, thermal emission and lidar (Jacob et al., 2016). The first solar backscattering mission was SCIAMACHY (Frankenberg et al., 2006), which was operational from 2003–2012 and observed the entire planet once every seven days. It was followed by GOSAT in 2009 (Kuze et al., 2016), and subsequently the next generation GOSAT-2 in 2018 (Glumb et al., 2014). In between, the TROPOMI mission was also launched in 2017, which observes the planet once daily with a high spatial resolution of  $7 \times 7$  km2 (Butz et al., 2012; Veefkind et al., 2012). CarbonSat (Buchwitz et al., 2013) is another proposed mission to measure CH4 globally from solar backscatter with a very fine spatial resolution  $(2 \times 2 \text{ km}^2)$  and high precision (0.4%). Thermal infrared observations of CH4 are available from the IMG (Clerbaux et al., 2003), AIRS (Xiong et al., 2008), TES (Worden et al., 2012), IASI (Xiong et al., 2013), and CrIS (Gambacorta et al., 2016) instruments. These instruments provide day/night measurements at spatial resolutions ranging from  $5\times8$  km2 (TES) to  $45\times45$  km2 (AIRS). GEO-CAPE (Fishman et al., 2012), GeoFTS (Xi et al., 2015), G3E (Butz et al., 2015), and GeoCarb (Polonsky et al., 2014) are proposed geostationary instruments (GeoCarb was selected by NASA under the Earth Venture - Mission program), which when operational will have resolutions of 2–5 km over regional scales. The MERLIN lidar instrument (Kiemle et al., 2014) scheduled for launch in 2021 will measure CH4 by employing a differential absorption lidar.

2. Line 171-172: How to do normalization for measured radiance? Add some description about this, please.

We have added some description to explain how and why the normalization is done (lines 197–199).

The normalization is done by calculating the ratio of the radiance to the maximum value across the spectral range, such that the values fall between 0 and 1. This is a first order correction for the effects of surface albedo.

3. Line 181: Is the typical XCH4 background of 1.822 ppm shown by the authors here related to the background covariance matrix and mean radiance used in MF method? Some reasons are expected here. By the way, it is better to mention the background covariance matrix and mean radiance in MF retrieval of CH4 plume case here.

For a real AVIRIS-NG measurement, the mean radiance and background covariance matrix used in the MF method are taken from a reference region close to the CH4 plume source. For the OE method in this case, a typical XCH4 background of 1.822 ppm is used. This typical value is obtained from annual mean data tabulated by the NOAA Global Monitoring Laboratory. For synthetic MF retrievals, we compute the mean radiance and background covariance matrix using simulations for the same typical XCH4 background of 1.822 ppm at different values of the surface albedo. We indicate how we obtained the typical value for the OE model in lines 208–211.

In the OE method, results are shown as a multiplicative scaling factor compared to a typical XCH4 background of 1.822 ppm. This value is the globally averaged marine surface annual mean for 2014 (Ed Dlugokencky, NOAA/GML (www.esrl.noaa.gov/gmd/ccgg/trends\_ch4/), the year corresponding to the AVIRIS-NG measurement being studied.

4. In the OE retrieval in section 3, what is the definition of the a priori value of XCH4? What aerosol model do the authors use? I think some parameters about aerosol model are expected here.

We have added the following description (lines 170–178) to provide the information the reviewer requested:

The *a priori* values are within 10% of the true values; *a priori* errors are assumed to be 20% for all state vector elements. The retrieved results are shown as the column averaged mixing ratio (XCH4, in ppm). Aerosols are not included in the state vector for both the real and synthetic retrievals. They are, however, considered in the forward model for the synthetic simulations. Table 1 lists optical properties for four basic aerosol types (dust, water soluble, oceanic and soot). Table 2 shows the corresponding properties for three aerosol models that are defined as mixtures of the basic components from Table 1 (WCRP, 1986). We employ the Henyey-Greenstein phase function (Henyey and Greenstein, 1941), where aerosol composition is determined by two parameters: single scattering albedo (SSA) and asymmetry parameter (g).

5. In section 4.3, the authors show the variation of OE XCH4 retrieval bias with SSA, g, AOD, surface albedo and XCH4. Which parameters affect XCH4 retrieval bias most? From aerosol parameters, which type of aerosols, such as smoke, dust or sea salt, causes the largest or lowest bias in XCH4 retrieval? These information will attract the audience's interest and provide guidance to correct aerosol impact in future XCH4 retrieval algorithm.

We have added the following sentences (lines 302-312) to describe the effects of SSA and g on the retrieval. We also added Figures 7b and 7c.

This behavior can be explained as follows. At higher SSA values, there are more multiple scattering effects (that are ignored in the retrieval). On the other hand, larger values of g imply greater anisotropy of scattering (preference for forward scattering), leading to a reduction in multiple scattering effects. Since the retrieval bias is large for high SSA and low g, the water-soluble aerosol type (Table 1) and the maritime aerosol model (Table 2) can be expected to induce greater biases in the retrieval. In order to compare the impacts of SSA and g in further detail, retrieval results due to a  $\pm$  5% change in SSA and g for the three aerosol models from Table 2 are shown in Figures 7b and 7c. Note that for the maritime aerosol model, the SSA is set to 0.999 for the +5% scenario to ensure physicality. It is clear that (1) the maritime aerosol model induces larger retrieval biases than the other aerosol types, and (2) the retrieval results are more sensitive to changes in g than those in SSA.

The AOD and surface albedo impacts are compared in lines 321–324.

The CH4 bias induced by differences in the surface albedo is not as large as that due to AOD variations, but albedo effects are noticeable at large AOD. Figure 8d shows the sensitivity of retrieval biases to changes in AOD and surface albedo, again demonstrating the greater impact of AOD than surface albedo in the retrieval.

6. In OE retrieval, the a priori error of XCH4 will affect the retrieval bias as well. Maybe the authors could check its impact.

The effect of changing the *a priori* error is described in lines 330–333.

Similarly, the XCH4 difference is less than 4 ppb when the *a priori* error changes from 0.05 to 0.5 (Figure 9b). Compared to the bias of about 923 ppb induced by neglecting aerosol scattering for this scenario, it is clear that the impacts of the *a priori* and *a priori* error are very small.

Technical corrections

1. Figure 9a and 9b have some overlaps with the same XCH4. There is no need to express them using two figures.

Our objective is to show two regimes, one where the OE method has lower bias ratio and the other where the MF method performs better. There is a crossover region between  $\sim 1.5$ –2 where both methods produce similar results. We believe that using two figures shows this behavior clearly. Some of the details might be lost if they were combined into one.

---

## Author Comment (AC2) · 24 Aug 2020

Dear Editor,

We would like to thank you and the two reviewers for your constructive comments and suggestions to improve the clarity of our manuscript. We have made changes to address these comments and suggestions. The following are the main changes:

1. More information has been added to the abstract to increase clarity, quantify the results better and summarize the comparison between the MF and OE methods.
2. Details have been added about the treatment of aerosols and surface albedo.
3. Explanations are supplied for the behavior of the retrievals as a function of the different parameters.
4. More quantification is provided for the differences between the two retrieval techniques.
5. Effects of changing the *a priori*, *a priori* error and simulation spectral resolution are described.
6. New Tables and Figures have been added to provide more detail.

Point-by-point responses to the comments are provided below. The reviewer comments are in blue, our responses are in red (line numbers refer to those in the revised manuscript), and modifications to the original manuscript are highlighted in yellow.

Vijay Natraj
On behalf of all co-authors

This paper studies the impact of aerosols on methane retrievals from synthetic AVIRIS-NG-like measurements using two retrieval methods, the traditional match filter (MF) method, and the more modern and quantitative optimal estimation (OE) method that uses radiative transfer model and can include more physics. It shows how both retrievals are sensitive to various AOD, CH4 concentration, surface albedo, SSA and g. The scope of this study is well suited for AMT. This paper is generally well organized and methodology is generally good. However, a lot of results are shown without much further explanation about the physics behind. Some of the statements may be questionable due to different definitions of retrieval biases between the two retrievals. Some key information (e.g., surface albedo treatment in OE retrieval) is not clearly described. The abstract also needs to be improved as it does not clearly summarize this study. Overall, I think that this paper can be published after addressing the specific comments below.

We thank the reviewer for the encouraging words and for the excellent suggestions. We have added/modified some sentences (lines 24–40) in the abstract to more clearly summarize the study.

Using a numerically efficient two-stream-exact-single-scattering radiative transfer model, we also simulate AVIRIS-NG measurements for different scenarios and quantify the impact of aerosol scattering in the two retrieval schemes by including aerosols in the simulations but not in the retrievals. The presence of aerosols causes an underestimation of $CH_4$ in both the MF and OE retrievals; the biases increase with increasing surface albedo and aerosol optical depth (AOD). Aerosol types with high single scattering albedo and low asymmetry parameter (such as water soluble aerosols) induce large biases in the retrieval. When scattering effects are neglected, the MF method exhibits lower fractional retrieval bias compared to the OE method at high $CH_4$ concentrations (2–5 times typical background values), and is suitable for detecting strong $CH_4$ emissions. For an AOD value of 0.3, the fractional biases of the MF retrievals are between 1.3 and 4.5%, while the corresponding values for OE retrievals are in the 2.8–5.6% range. On the other hand, the OE method is an optimal technique for diffuse sources (<1.5 times typical background values), showing up to five times smaller fractional retrieval bias (8.6%) than the MF method (42.6%) for the same AOD scenario. However, when aerosol scattering is significant, the OE method is superior since it provides a means to reduce biases by simultaneously retrieving AOD, surface albedo and $CH_4$. The results indicate that, while the MF method is good for plume detection, the OE method should be employed to quantify $CH_4$ concentrations, especially in the presence of aerosol scattering.

The treatment of aerosols and surface albedo is now described in lines 172–179.

Aerosols are not included in the state vector for both the real and synthetic retrievals. They are, however, considered in the forward model for the synthetic simulations. Table 1 lists optical properties for four basic aerosol types (dust, water soluble, oceanic and soot). Table 2 shows the corresponding properties for three aerosol models that are defined as mixtures of the basic components from Table 1 (WCRP, 1986). We employ the Henyey-Greenstein phase function (Henyey and Greenstein, 1941), where aerosol composition is determined by two parameters: single scattering albedo (SSA) and asymmetry parameter ($g$). The surface albedo is also not retrieved; for both real and synthetic retrievals, it is held fixed and assumed to be independent of wavelength.

Further description of the retrieval bias as a function of SSA and $g$ is provided in lines 302–312.

This behavior can be explained as follows. At higher SSA values, there are more multiple scattering effects (that are ignored in the retrieval). On the other hand, larger values of $g$ imply greater anisotropy of scattering (preference for forward scattering), leading to a reduction in multiple scattering effects. Since the retrieval bias is large for high SSA and low $g$, the water-soluble aerosol type (Table 1) and the maritime aerosol model (Table 2) can be expected to induce greater biases in the retrieval. In order to compare the impacts

of SSA and $g$ in further detail, retrieval results due to a ± 5% change in SSA and $g$ for the three aerosol models from Table 2 are shown in Figures 7b and 7c. Note that for the maritime aerosol model, the SSA is set to 0.999 for the +5% scenario to ensure physicality. It is clear that (1) the maritime aerosol model induces larger retrieval biases than the other aerosol types, and (2) the retrieval results are more sensitive to changes in $g$ than those in SSA.

The effects of changing the *a priori*, *a priori* error and RT simulation spectral resolution are described in lines 325–334.

The effects of changing the *a priori*, *a priori* error and RT simulation spectral resolution on the retrieved $XCH_4$ are shown in Figure 9. For these calculations, the other parameters are set as follows: SSA = 0.95, $g$ = 0.75, AOD = 1.0, surface albedo = 0.5 and true $XCH_4$ = 5.8 × 1.822ppm. The parameters were chosen to correspond to the scenario with the largest retrieval bias in Figure 8c (bottom right box in Figure 8c). Figure 9a shows that the retrieved $XCH_4$ changes by about 9 ppb as the *a priori* changes from half to twice the true $XCH_4$ value. Similarly, the $XCH_4$ difference is less than 4 ppb when the *a priori* error changes from 0.05 to 0.5 (Figure 9b). Compared to the bias of about 923 ppb induced by neglecting aerosol scattering for this scenario, it is clear that the impacts of the *a priori* and *a priori* error are very small. The effect of spectral resolution is larger, but $XCH_4$ still changes by only about 100 ppb when the spectral resolution is changed from 0.5 to 0.1 cm$^{-1}$ (Figure 9c).

We provide more quantification of the differences between the two retrieval techniques (lines 339–351).

The bias ratio for the MF method (1.3–4.5%) is up to 53.6% less than that for the OE method (2.8–5.6%) for AOD = 0.3 when the $CH_4$ concentration is high (2–5 times typical background values). On the other hand, the OE method performs better when enhancements are small and $XCH_4$ is close to the background value. For example, the bias ratio for the MF method has a high value of about 42.6% at AOD = 0.3 for a 10% enhancement ($XCH_4$ = 1.1 × 1.822 ppm); the OE value for the same scenario is 8.6%. For scenarios where scattering is ignored, the two retrieval techniques seem to be complementary, with differing utilities for different enhancements. On the other hand, when RT models that account for scattering effects are employed, the MF technique is suboptimal. Further, MF retrievals rely on accurate characterization of the surface albedo, especially when the aerosol loading is large. Finally, the MF method does not retrieve concentrations, which are necessary to infer fluxes. Therefore, the OE technique is in general superior due to its ability to support simultaneous retrieval of aerosols, surface albedo and $CH_4$ concentration.

Finally, we have added two new Tables (Tables 1 and 2) and several new Figures (Figures 7b, 7c, 8d and 9).

Specific comments
1. Subscripts and superscripts in the text are disproportionally too small to read.

The subscripts and superscripts look fine in our version. Perhaps this is an issue that could be addressed by the journal publication team after acceptance.

2. L19-21, this sentence is not consistent with the text as real AVIRIS-NG data are mainly used to compare both MF and OE retrievals, rather than analyze the impact of aerosol scattering on CH4 retrievals. That is probably why the first reviewer commented that section 3 is loosely connected with the main purpose of this paper. I suggest clearly describing the purpose of this section 3 probably in the introduction section in relation to the main topics of this study.

The reviewer is right. We modified the abstract to indicate the use of AVIRIS-NG data (lines 21–24).

In this study, imaging spectroscopic measurements from the Airborne Visible/Infrared Imaging Spectrometer–Next Generation (AVIRIS-NG) in the short-wave infrared are used to compare two retrieval techniques — the traditional Matched Filter (MF) method and the Optimal Estimation (OE) method, which is a popular approach for trace gas retrievals.

We also describe the organization of this work at the end of the introduction (lines 113–118).

This article is organized as follows. The MF and OE retrieval methods are described in Section 2. Section 3 focuses on analysis of a sample $CH_4$ plume detected by AVIRIS-NG measurements and compares retrievals using the MF and OE methods. Section 4 presents a detailed evaluation of aerosol impacts on the two retrieval methods through simulations of AVIRIS-NG spectra for different geophysical parameters. Section 5 provides a summary of the work and discusses future research.

3. In abstract, the sentence in L25-29, it is not clear about what kind of scenario for the retrieval bias. Please make it clear here that this for retrieval underestimation of CH4 when aerosol is present but neglected in the retrieval. The use of 50% here and also in the text in Section 4.4 is very confusing especially you have >100% enhancement and < 50% enhancement and also the retrieval bias is actually on the order of 1.5-6%. You may use

something like "half of the retrieval bias" or provide specific retrieval bias (~2-6% for OE and 1,5-3.5% for MF) based on Figure 9. The sentence is also too long. You may rephrase it in a couple of sentences.

We thank the reviewer for the excellent suggestions. We now indicate that aerosols are present in the simulations but neglected in the retrieval (lines 24–27).

Using a numerically efficient two-stream-exact-single-scattering radiative transfer model, we also simulate AVIRIS-NG measurements for different scenarios and quantify the impact of aerosol scattering in the two retrieval schemes by including aerosols in the simulations but not in the retrievals.

We have also rephrased the sentences comparing the two retrievals (lines 30–37).

When scattering effects are neglected, the MF method exhibits lower fractional retrieval bias compared to the OE method at high $CH_4$ concentrations (2–5 times typical background values), and is suitable for detecting strong $CH_4$ emissions. For an aerosol optical depth (AOD) value of 0.3, the fractional biases of the MF retrievals are between 1.3 and 4.5%, while the corresponding values for OE retrievals are in the 2.8–5.6% range. On the other hand, the OE method is an optimal technique for diffuse sources (<1.5 times typical background values), showing up to five times smaller fractional retrieval bias (8.6%) than the MF method (42.6%) for the same AOD scenario.

We add two sentences to summarize the implications of this work (lines 37–40).

However, when aerosol scattering is significant, the OE method is superior since it provides a means to reduce biases by simultaneously retrieving AOD, surface albedo and $CH_4$. The results indicate that, while the MF method is good for plume detection, the OE method should be employed to quantify $CH_4$ concentrations, especially in the presence of aerosol scattering.

We have also provided further details in Section 4.4 (lines 339–351).

The bias ratio for the MF method (1.3–4.5%, AOD = 0.3) is up to 53.6% less than that for the OE method (2.8–5.6%, AOD = 0.3) when the $CH_4$ concentration is high (2–5 times typical background values). On the other hand, the OE method performs better when enhancements are small and $XCH_4$ is close to the background value. For example, the bias ratio for the MF method has a high value of about 42.6% at AOD = 0.3 for a 10% enhancement ($XCH_4 = 1.1 \times 1.822$ ppm); the OE value for the same scenario is 8.6%. For scenarios where scattering is ignored, the two retrieval techniques seem to be

complementary, with differing utilities for different enhancements. On the other hand, when RT models that account for scattering effects are employed, the MF technique is suboptimal. Further, MF retrievals rely on accurate characterization of the surface albedo, especially when the aerosol loading is large. Finally, the MF method does not retrieve concentrations, which are necessary to infer fluxes. For these cases, the OE technique is superior due to its ability to support simultaneous retrieval of aerosols, surface albedo and $CH_4$ concentration.

4. In abstract, L29-31, it is good to summarize main results instead of just describing what are discussed.

We now summarize the main results per the reviewer's suggestion (lines 27–37).

The presence of aerosols causes an underestimation of $CH_4$ in both the MF and OE retrievals; the biases increase with increasing surface albedo and aerosol optical depth (AOD). Aerosol types with high single scattering albedo and low asymmetry parameter (such as water soluble aerosols) induce large biases in the retrieval. When scattering effects are neglected, the MF method exhibits lower fractional retrieval bias compared to the OE method at high $CH_4$ concentrations (2–5 times typical background values), and is suitable for detecting strong $CH_4$ emissions. For an AOD value of 0.3, the fractional biases of the MF retrievals are between 1.3 and 4.5%, while the corresponding values for OE retrievals are in the 2.8–5.6% range. On the other hand, the OE method is an optimal technique for diffuse sources (<1.5 times typical background values), showing up to five times smaller fractional retrieval bias (8.6%) than the MF method (42.6%) for the same AOD scenario.

5. L74, suggest changing "large" to "large number of " as it implies coarser spatial resolution contrary to "fine spatial resolution".

We have made the suggested revision (line 85).

6. L78, suggest changing to "a spectral resolution of 5 nm full width at half maximum (FWHM)"

We have made the suggested revision (line 89).

7. At the end of the introduction, it would be useful to add how this paper is organized in following sections.

We now describe the organization of the paper (lines 113–118).

This article is organized as follows. The MF and OE retrieval methods are described in Section 2. Section 3 focuses on analysis of a sample $CH_4$ plume detected by AVIRIS-NG measurements and compares retrievals using the MF and OE methods. Section 4 presents a detailed evaluation of aerosol impacts on the two retrieval methods through simulations of AVIRIS-NG spectra for different geophysical parameters. Section 5 provides a summary of the work and discusses future research.

8. Units on both sides of Equation (5) do no match. According to the text, V has a unit of liter / mol or IE-3 m"3/mol, and 1/(V*1E3) has a unit of mol/m"(-3). Maybe 1/(V*1E3) should be V*1E-3 instead. Or V has a unit of mol/liter, then it should not be called V as it is confusing. So please clarify this.

Good catch. Not only was the equation wrong, but it was also confusing. The equation now reads (line 147):

$$\kappa_{trad} \, [\text{m}^2 \cdot \text{mol}^{-1}] = \kappa \, [\text{ppm}^{-1}\text{m}^{-1}] \times V \, [\text{liter mol}^{-1}] \times 10^{-3} \, [\text{m}^3 \, \text{liter}^{-1}] \, / \, 10^{-6} \, [\text{ppm}^{-1}] \quad (5)$$

$\kappa_{trad}$ is defined in line 142 as the absorption coefficient in units of $\text{m}^2 \cdot \text{mol}^{-1}$.

9. L132-135, the sentence does not read well here as the purpose of using real AVIRIS-NG data has not been introduced yet. You may rephrase it to something like "To illustrate the MF retrieval and its difference from the OE method, we perform MF retrievals from AVIRIS-NG measurement made on ... as shown in Fig. 2. The samples for..." Or it might be even better to move these two sentences to Section 3 before showing MF retrieval results.

The reviewer is right. We modified the first sentence (lines 149–150).

The background mean radiance $\mu$ used in Equation 4 is based on the AVIRIS-NG measurement shown in Figure 2; this is described in more detail in Section 3.

As recommended by the reviewer, we added a sentence at the start of Section 3 (lines 182–184) to introduce the purpose of using AVIRIS-NG data.

To illustrate the OE retrieval and its difference from the MF method, we perform retrievals for an AVIRIS-NG measurement made on 4 September 2014 (ang20140904t204546) in Bakersfield, CA, as shown in Figure 2.

We moved the sentence about the background covariance matrix to later in Section 3 (lines 205–208).

In the MF method, the background covariance matrix $\Sigma$ and mean radiance $\mu$ are drawn from a reference region close to the $CH_4$ emission source. These are shown in Figure 2, where the dashed green box denotes the reference region and the source is located within the solid red box.

10. L167, it might be good to describe some of the retrieval artifacts and why they are produced. Are some of the retrieval artifacts related to aerosols or surface albedo?

We have added a description of the artifacts (lines 191–193).

Some artifacts caused by surfaces with strong absorption in the 2100–2500 nm wavelength range, such as oil-based paints or roofs with calcite as a component (Thorpe et al., 2013), also produce large $\alpha$ values in the MF method

11. L169-172, the first sentence seems to be redundant with previous description and can be removed. Also good to describe how the normalization is done and its main purpose.

We have removed the first sentence. and added some description to explain how and why the normalization is done (lines 197–199).

The normalization is done by calculating the ratio of the radiance to the maximum value across the spectral range, such that the values fall between 0 and 1. This is a first order correction for the effects of surface albedo.

12. L173-174, the sentence "The radiance has units can be removed as the spectral range has already been mentioned earlier in the paragraph and it is not necessary to mention the units of radiance.

We have deleted the sentence.

13. L206, the spectral resolution of 0.5cm"-1 seems to be too coarse to resolve monochromatic spectral features in this spectral region. Have you performed sensitivity calculation to see how this affect the synthetic AVIRIS-NG radiance?

AVIRIS-NG has 400 channels overall, and 80 within the 2100–2500 nm range. The spectral resolution of 0.5 cm$^{-1}$ provides more than 1520 monochromatic wavelengths. Therefore, there are around 20 monochromatic calculations per spectral channel, which should be sufficient. Nevertheless, we performed a sensitivity calculation as recommended by the reviewer. Changing the spectral resolution to 0.1 cm$^{-1}$ resulted in ~1% difference in the retrieved XCH$_4$. We have added Figure 9c to show the effect of spectral resolution on the retrieval.

14. L178-186, it is good to mention clearly whether aerosol is included in both the forward model and retrieval. It seems that aerosol is not retrieved, but not sure if fixed aerosol model is used in the forward model as it mentions "Single scattering ... using all moments of the phase function"

For the AVIRIS-NG retrievals, aerosols are neither included in the forward model nor in the retrieval. This is now clearly mentioned in Section 2.2 (lines 172–173).

Aerosols are not included in the state vector for both the real and synthetic retrievals. They are, however, considered in the forward model for the synthetic simulations.

This point is reiterated in Section 3 (line 217).

Aerosols are neither included in the forward model nor retrieved in this analysis.

15. L187-191, although H2O is not retrieved and taken into account in the MF method, it should also cause retrieval bias/uncertainty to the MF result. Probably it will cause different retrieval errors to MF method and OE method due to its different retrieval treatments. Also are some of the differences due to aerosols and surface albedo?

In the MF method, the background mean radiance and covariance matrix are drawn from a region near the plume. We assume that the region used to compute the background radiance has the same H$_2$O concentration as the plume. However, in the OE method, we retrieve H$_2$O and CH$_4$ simultaneously. These gases have overlapping absorption in the retrieval wavelengths. Note that we normalize the radiance to reduce the influence of surface albedo. Further, aerosols are not considered in either retrieval. Therefore, it is likely that the difference in the treatment of H$_2$O is the main source of the discrepancy in the retrievals. We have modified the following sentence to clarify this point (lines 221–224).

==Since radiance normalization reduces the impact of surface albedo and aerosols are not included in either retrieval, this might be due to the fact that, in the OE method, $H_2O$ and $CH_4$ are simultaneously retrieved; the $CH_4$ retrieval has added uncertainty due to overlapping absorption features between these two gases.==

16. L226, the absorption cross-section is independent of concentration, suggest removing "cross-section"

We have made the suggested change.

17. L228, why does the effect of aerosol loading cause underestimation? Would be good to provide some explanation. Due to the shielding of CH4 absorption below aerosol layer?

The reviewer is right. The effect of aerosol loading is to shield $CH_4$ absorption below the aerosol layer. There is also another effect — it increases the path length (due to multiple scattering between the aerosol layer and the surface). When aerosol scattering is ignored in the retrieval, it manifests as reduced $CH_4$ absorption. We add a brief description per the reviewer's suggestion (lines 265–267).

==The underestimation, which is due to the shielding of $CH_4$ absorption below the aerosol layer and the fact that multiple scattering effects between the aerosol and the surface are ignored, is clearly shown in Figure 5b==

18. L232-233 and in Fig. 5c, are the results really independent of surface albedo here? Or is this simply because a background with the same surface albedo is used? In actual MF retrievals, surface albedo is not necessarily known (or be the same as that in the background). Also according to normalization procedure shown in Fig. 2, looks like most of the surface albedo can be taken into account after the normalization if surface albedo is not perfectly known as in real retrievals, but bias will occur. Please clarify this.

We assume different surface albedos to calculate the covariance matrix and mean background radiance for the MF method. We did calculate retrieval biases for different surface albedos, but the results are indistinguishable. Hence, we just include one line in Figure 5c. The reason for this behavior is that when there is no aerosol loading, there are no multiple scattering effects between the surface and the atmosphere (Rayleigh scattering is negligible in the retrieval wavelength range). We add the following sentence (lines 271–

273) to clarify this point.

This is because there are no multiple scattering effects between the surface and the atmosphere (Rayleigh scattering is negligible in the retrieval wavelength range) when there is no aerosol loading.

The reviewer's point is well taken, though. The results clearly show that the enhancement depends on the surface albedo when AOD > 0. The spread increases as the AOD becomes larger. We add another sentence (lines 279–280) to point that out.

The implication of these results is that accurate knowledge of the surface albedo is important for MF retrievals, especially when the aerosol loading is large.

19. L236, good to explain why larger biases at large AOD and surface albedo values. Also since the enhancement in units of ppm m is retrieved with the MF method, it is better to mention the bias in enhancement (∼-700 ppm m) rather than saying "maximum bias ..

close to 0.06 x 1.822 ppm), or you can say "the maximum bias is close to ∼-700 ppm m (equivalent of -0.06 x 1.822 ppm) ..."

We have implemented the reviewer's suggestion and added further description of the physical reasoning behind the larger biases (lines 273–279).

For the scenarios with aerosol loading, the dispersion in the zero-enhancement $XCH_4$ value between different surface albedos indicates that results from the MF method are biased more at large AOD and surface albedo values (Figures 5c–f). This is a consequence of increased multiple scattering between the aerosol layer and the surface that is not accounted for by the retrieval algorithm. The maximum bias value is close to $-700$ ppm $\times$ m (equivalent to $-0.06 \times 1.822$ ppm relative to the background concentration of $1.0 \times 1.822$ ppm) for an AOD of 0.3 and albedo of 0.5 (Figure 5f).

20. L242-243, why does the bias decreases with increasing CH4 concentration for the MF method? The reason givens on L244-245 only shows that the enhancement is more underestimated at larger $XCH_4$ concentration (as shown from the curves in Figure 5a that

deviated from a straight line), and seems not able to explain the enhancement difference between without and with aerosols decreases with increasing CH4 concentration.

The reviewer is right. The enhancement underestimation at large $XCH_4$ values is shown by the deviation of Figure 5a from a straight line. The enhancement bias trend is a result of the changing slope of this curve as a function of $XCH_4$, which in turn is due to the increasing inaccuracy of the linear MF method at large XCH4 values, where the absorption behavior is highly nonlinear. We explain this in lines 288–292.

This surprising behavior is a direct consequence of the physical basis of the MF method. The rate of increase in enhancement becomes smaller as $XCH_4$ becomes larger (Figure 5a). Therefore, at higher $XCH_4$ values, the addition of aerosols (which has a similar effect as a reduction in $XCH_4$) results in a lower reduction in enhancement compared to that at lower $XCH_4$ values, resulting in a net decrease in the enhancement bias.

21. Figure 6, the bias is negative (underestimated) as indicated in the text. Suggest making it clear in Fig. 6 caption that the figure shows the magnitude of underestimation.

We apologize for the confusing definition. The bias is defined as the difference between the enhancement values without and with aerosol (i.e. without-with). We clarify this in the revised manuscript (lines 282–284).

The color bar shows the $\alpha$ bias — which is defined as the difference between the enhancement value without aerosol (true $\alpha$ value) and that with aerosol — for different CH4 concentrations, surface albedos and AODs. A positive bias means that CH4 is underestimated.

22. L255 and Figures 7,8,9, is the bias also negative? If so, please make it clear.

The bias here is defined as the difference between the true and retrieved $XCH_4$. Therefore, the bias is positive. We clarify this in the revised manuscript (lines 297–298).

the retrieval bias is defined as the difference between the true $XCH_4$ in the simulation and the retrieved value (positive values refer to underestimation).

23. L258, good to explain how the bias varies with SSA and g.

We add the following sentences (lines 302–305) to explain the bias variation with SSA and *g*.

This behavior can be explained as follows. At higher SSA values, there are more multiple scattering effects (that are ignored in the retrieval). On the other hand, larger values of g imply greater anisotropy of scattering (preference for forward scattering), leading to a reduction in multiple scattering effects.

24. L259-267 and results in Figure 8: is surface albedo retrieved? It seems to me that it is retrieved with XCH4 so that the error is small for different surface albedo when AOD=0. While for other cases (e.g., surface albedo is kept constant at 0.3), maybe surface albedo is not retrieved. Please make it clear probably at the end of section 2.2 or in this paragraph about whether surface albedo is retrieved and how it is retrieved (e.g., wavelength independent or dependent)

Please note that simulations for this figure are done for AOD between 0.1 and 1. In any case, for AOD = 0, there are no multiple scattering effects between the aerosol and the surface and hence no uncertainly in path length. This is why the error is small for different surface albedos for this scenario. The surface albedo is not retrieved. In the simulations, we set it to a wavelength-independent value. We describe this in Section 2.2 (lines 178–179) and in Section 4.3 (lines 295–296).

The surface albedo is also not retrieved; for both real and synthetic retrievals, it is held fixed and assumed to be independent of wavelength.

The OE method is then used to perform retrievals using the same configuration (including, in particular, the same surface albedo) except that AOD is set to zero.

25. L264, it seems to me that the bias is defined differently for the MF case, as the enhancement between without and with aerosols (L239-240), while the bias for OE is defined as the difference between retrieved and true XCH4. If we use a similar definition, according to Figure 5a, there is larger underestimation at higher XCH4 values for both without and with aerosols in the MF method.

It is not possible to use similar definitions for the two methods. The OE method retrieves

XCH$_4$, while the MF method retrieves the enhancement. Further, Figure 5a shows the enhancement as a function of XCH$_4$. The enhancement increases with XCH$_4$. That is not the same thing as saying that the XCH$_4$ biases are larger. The enhancement bias (plotted in Figure 6) is a better representation of the effects of aerosol scattering.

26. L273, please check if it should be between "with and without aerosols" as the case with zero AOD is the truth reference for the MF method.

The reviewer is right. We have rephrased the sentence as follows (lines 337–338).

On the other hand, in the OE method, it is the ratio of the bias to the true XCH$_4$.

27. L275-281, how is this OE retrieval sensitive to the assumed a priori error of 20%? If you use a larger a priori error for the OE method, will the conclusion here be changed?

The effect of changing the *a priori* error is described in lines 330–333.

Similarly, the XCH$_4$ difference is less than 4 ppb when the *a priori* error changes from 0.05 to 0.5 (Figure 9b). Compared to the bias of about 923 ppb induced by neglecting aerosol scattering for this scenario, it is clear that the impacts of the *a priori* and *a priori* error are very small.

28. L280, an example is given for a XCH4 of 1.1x1.822ppm. It is useful to give another example at high XCH4, for example XCH4=5.0.

We provide bias ratio comparisons for a range of scenarios (lines 339–341).

The bias ratio for the MF method (1.3–4.5%) is up to 53.6% less than that for the OE method (2.8–5.6%) for AOD = 0.3 when the CH$_4$ concentration is high (2–5 times typical background values).

29. Section 5 is a summary of this paper and discussion about future work, I suggest changing this section title to "Summary and discussion"

We have made the suggested change.

30. L289-291, the sentence might not be true as mentioned earlier due to different bias definitions used for OE and MF methods.

The bias definitions are different by necessity, as described in the response to comment #25. Within the framework of those definitions, the statement holds.

---

## Author Response (AR2)

Dear Editor,

We would like to thank you and the reviewer for your constructive comments and suggestions to improve the clarity of our manuscript. We have made changes to address these comments and suggestions.

Point-by-point responses to the comments are provided below. The reviewer comments are in blue, our responses are in red (line numbers refer to those in the revised manuscript), and modifications to the original manuscript are highlighted in yellow.

Please also note that we added an extra author (Pushkar Kopparla) in the second round of revisions. That has not been reflected in the manuscript website.

Vijay Natraj
On behalf of all co-authors

Some of the approved satellite missions (e.g., GHGSAT-D, MethaneSAT) to measure $CH_4$ are not mentioned while other proposed missions (proposal submitted but not funded or concept published) were mentioned. So you may add some of these on-going missions: GHGSAT-D (McKeever, et al., 2017, AGU Fall Meeting, Varon et al., 2019, GRL), MethaneSAT (Wofsy and Hamburg, 2019, AGU Fall meeting).

We agree with the reviewer and have added the following sentences (lines 76–80).

[revised manuscript text omitted]